# Statistical Analysis Methods Applied to Early Outpatient COVID-19 Treatment Case Series Data

**Eleftherios Gkioulekas [1,\*], Peter A. McCullough [2] and Vladimir Zelenko [3,†]**

1   School of Mathematical and Statistical Sciences, University of Texas Rio Grande Valley, Edinburg, TX 78539, USA
2   Truth for Health Foundation, Tucson, AZ 85728, USA
3   Former Affiliate Physician, Columbia University Irving Medical Center, New York, NY 10032, USA
\*   Correspondence: drlf@hushmail.com
†   Vladimir Zelenko passed away on 30 June 2022. We dedicate this paper to his memory.

**Abstract:** When confronted with a public health emergency, significant innovative treatment protocols can sometimes be discovered by medical doctors at the front lines based on repurposed medications. We propose a statistical framework for analyzing the case series of patients treated with such new protocols, that enables a comparison with our prior knowledge of expected outcomes, in the absence of treatment. The goal of the proposed methodology is not to provide a precise measurement of treatment efficacy, but to establish the existence of treatment efficacy, in order to facilitate the binary decision of whether the treatment protocol should be adopted on an emergency basis. The methodology consists of a frequentist component that compares a treatment group against the probability of an adverse outcome in the absence of treatment, and calculates an efficacy threshold that has to be exceeded by this probability, in order to control the corresponding *p*-value and reject the null hypothesis. The efficacy threshold is further adjusted with a Bayesian technique, in order to also control the false positive rate. A random selection bias threshold is then calculated from the efficacy threshold to control for random selection bias. Exceeding the efficacy threshold establishes the existence of treatment efficacy by the preponderance of evidence, and exceeding the more demanding random selection bias threshold establishes the existence of treatment efficacy by the clear and convincing evidentiary standard. The combined techniques are applied to case series of high-risk COVID-19 outpatients that were treated using the early Zelenko protocol and the more enhanced McCullough protocol.

**Keywords:** COVID-19; SARS-CoV-2; ambulatory treatment; early treatment; mortality; hospitalization; epidemiology; biostatistics; drug repurposing



## 1. Introduction

In medical research, the efficacy of new drugs or treatment protocols is established by controlled studies in which a treatment group is compared against a control group. A case series is one half of a controlled study consisting only of the treatment group. At the beginning of the COVID-19 pandemic, practicing medical doctors were confronted with having no treatment to offer to their patients that can prevent or minimize hospitalization and/or death. In response, some doctors were compelled to innovate and discover, on their own, treatment protocols using repurposed off-label medications. Most notable examples, amongst several others, include Didier Raoult [1] in the  IHU Méditerranée Infection hospital in Marseilles France, Vladimir Zelenko [2] in upstate New York, George Fareed and Brian Tyson [3] in California, Shankara Chetty [4] in South Africa, Jackie Stone [5] in Zimbabwe, and Paul Marik's group [6,7], which was in the beginning based at the Eastern Virginia Medical School. Their efforts to treat patients generated case series of successfully treated patients that constitute real-world evidence [8].

The goal of this paper is to present a statistical framework for rapidly analyzing systematic case series data of early treatment protocols with binary endpoints (e.g., hospitalization or death), and comparing them against our prior knowledge of the likelihood of adverse outcomes in the absence of treatment. Although the development of the proposed statistical technique was originally motivated by the need to assess available case series [2,9–13] of multidrug treatment protocols [2,14–16] for COVID-19, it can also play a very important role in the public health response to future pandemics or epidemics with no established treatment protocols. Furthermore, the potential scope of our methodology is very broad and it can be used to compare any treatment group case series, with binary endpoints, against our prior knowledge of the probability of adverse outcomes based on population-level historical controls. A limitation of the methodology is that it should be used only for treatment protocols that are based on repurposed medications [17] with known acceptable safety. The main advantage of the technique is that it can be very good at rapidly validating and enabling the deployment of treatment protocols, based on repurposed medications, when there is a sufficiently strong signal of efficacy. When confronted with a mass casualty event, it is critically important to be able to rapidly leverage the direct clinical experience of medical doctors, towards formulating an evidence-based standard of care, while also being able to rigorously quantify the quality of the available evidence.

The closest concept to our approach is the idea of using a virtual control group [18], in which the outcomes observed in a treatment group case series are compared against the predicted outcomes for the same patient cohort without treatment, using a trained statistical model, based on data accumulated before the discovery of the treatment in question. The virtual control group method aims to not only establish the existence of efficacy, but to also measure the corresponding treatment efficacy. Our idea is to abandon any attempt to obtain an unbiased measure of the treatment efficacy, and to focus on establishing, with sufficient confidence, the existence of some positive treatment efficacy. We do this by comparing the treatment group case series with a probability lower-bound for the expected negative outcomes without treatment. Such lower bounds can be easily extracted from available data, and can be facilitated by applying risk-stratification on the treatment group case series, when necessary. Thus, our aim is to establish, with sufficient confidence, a positive lower bound for treatment efficacy, quickly and without expending substantial resources, using real-world evidence that has been accumulated from the efforts of practicing physicians. In turn, this can be sufficient for a positive recommendation to adopt the corresponding treatment protocol.

Because case series are susceptible to selection bias, we define two cross-over thresholds for making a positive recommendation: an efficacy threshold, corresponding to a *preponderance of evidence* standard, where we assume there is no selection bias, and a random selection bias threshold, corresponding to the *clear and convincing* evidentiary standard, which controls for random selection bias in the case series. Following the recommendation of the American Statistical Association statement on statistical significance and *p*-values [19], the proposed approach combines use of the *p*-value, which enables one to reject the null hypothesis, with a Bayesian factor analysis framework [20–24] for controlling the false positive rate [25] in the calculation of the efficacy threshold. Empirically, we have found that the frequentist p-value framework has done a pretty good job on its own, at least for the analysis of the case series data considered in this paper. However, complementing it with Bayesian factor analysis is a reasonable precaution, because it can help raise the red flag when dealing with small sample sizes and/or weak signals.

We apply the proposed framework to the processing of available case series data [2,9–13] that support proposed early outpatient treatment protocols for COVID-19 patients, such as the original Zelenko triple-drug protocol [2] and the more advanced McCullough protocol [14–16]. The original Zelenko protocol was first announced on 23 March 2020 [26]. The proposed approach was to risk stratify patients into two groups (low-risk vs. high-risk), provide supportive care to the low-risk group, and treat the high-risk group with a triple-drug protocol (hydroxychloroquine, azithromycin, zinc sulfate).

Results were reported in an 28 April 2020 letter [9] and a 14 June 2020 letter [10], and the lab-confirmed subset of the April data was published in a formal case-control study [2]. Zelenko's letters have been attached to our supplementary material document [27].

The rationale for the triple-drug therapy was based on the following mechanisms of action: hydroxychloroquine prevents the virus from binding with the cells, and also acts as a zinc ionophore that brings the zinc ions inside the cells, which in turn inhibit the RDRP (RNA dependent RNA polymerase) enzyme used by the virus to replicate [28,29]. Azithromycin's role is to guard against a secondary infection, but we have since learned that it also has its own anti-viral properties [30–32], and a signal of the efficacy of adding azithromycin on top of hydroxychloroquine can be clearly discerned in a study of nursing home patients in Andorra, Spain [33].

It is interesting that chloroquine was shown in vitro to have antiviral properties against the previous SARS-CoV-1 virus [34], and that there is an anecdotal report from 1918 [35] about the successful use of quinine dihydrochloride injections as an early treatment of the Spanish flu. In hindsight, it is now known that influenza viruses also use the RDRP protein to replicate [36], which can be inhibited with intracellular zinc ions [28,29]. Consequently, there is a mechanism of action that can explain why we should anticipate the combination of zinc with a zinc ionophore (i.e., hydroxychloroquine, or quercetin [37], or EGCG (Epigallocatechin Gallate) [38]) to inhibit the replication of the influenza viruses. Other RNA viruses, including the respiratory syncytial virus (hereafter RSV) [39] and the highly pathogenic Marburg and Ebola viruses [40,41], are also using the RDRP protein to replicate, raising the question of whether the zinc/zinc ionophore concept could also play a useful role against them.

Zelenko's protocol was soon extended into a sequenced multidrug approach, known as the McCullough protocol [14–16], which is based on the insight that COVID-19 is a tri-phasic illness that manifests in three phases: (1) an initial viral replication phase, in which the virus infects cells and uses them to replicate and make new viral particles, during which patients present with flu-like symptoms; (2) an inflammatory hyper-dysregulated immune-modulatory florid pneumonia, that presents with a cytokine storm, coughing, and shortness of breath, triggered by the toxicity of the spike protein [42], when it is released, as viral particles are destroyed by the immune system, triggering release of interleukin-6 and a wave of cytokines; (3) a thromboembolic phase, during which microscopic blood clots develop in the lungs and the vascular system, causing oxygen desaturation, and very damaging complications that can include embolic stroke, deep vein thrombosis, pulmonary embolism, myocardial injury, heart attacks, and damage to other organs.

The rationale of the original Zelenko protocol was that early intervention to stop the initial viral replication phase could prevent the disease from progressing to the second and third phase, and, in doing so, prevent hospitalizations or death. The McCullough protocol [14–16] extends the Zelenko protocol by using multiple drugs in combination sequentially to mitigate each of the three phases of the illness, depending on how they present for each individual patient. McCullough's therapeutic recommendations for handling the cytokine injury phase and the thrombosis phase of the COVID-19 illness are, for the most part, standard on-label treatments for treating hyper-inflammation and preventing blood clots. The most noteworthy innovations to the antiviral part of the protocol are the addition of ivermectin [43–48], which has 20 known mechanisms of action against COVID-19 [49], to be used as an alternative or in conjunction with hydroxychloroquine, the addition of a nutraceutical bundle [50–52] combined with a zinc ionophore (quercetin [37] or EGCG [38]) for both low-risk and high-risk patients, and lowering the age threshold for high-risk patients to 50 years. The MATH+ protocol [6,7], developed for hospitalized patients by Marik's group, follows the same principles of a sequenced multidrug treatment. A similar treatment protocol, based on similar insights, was independently discovered and published on May 2020 by Chetty [4] in South Africa.

McCullough's protocol [14–16] was adopted by some treatment centers throughout the United States and overseas, but has not been endorsed by the United States public health

agencies, ostensibly due to lack of support of the entire sequenced treatment algorithm by an RCT (Randomized Controlled Trial). In spite of the urgent need for safe and effective early outpatient treatment protocols for COVID-19, there has been no attempt to conduct any such trials of any comprehensive multidrug outpatient treatment protocols throughout the pandemic. Instead, the prevailing approach has been to try to build treatment protocols, one drug at a time, after validating each drug with an RCT. Because COVID-19 is a multifaceted tri-phasic illness, there is no a priori reason to expect that a single drug alone will work for all three phases of the disease. Consequently, the first priority should be to validate the efficacy of treatment protocols that use multiple drugs in combination, since this is what is actually going to be used in practice to treat patients. To that end we have analyzed the case series by Zelenko [2,9,10], Procter [11,12], and Raoult [13], where such multidrug outpatient treatment protocols have been used by practicing physicians.

The broader context in which the proposed statistical methodology is situated is as follows. Shortly before COVID-19 was declared a pandemic by the World Health Organization, an article [53] was published on 23 February 2020 in the New England Journal of Medicine arguing that *"the replacement of randomized trials with non-randomized observational status is a false solution to the serious problem of ensuring that patients receive treatments that are both safe and effective."* The opposing viewpoint was published earlier in 2017 by Frieden [54], highlighting the limitations of RCTs and the need to leverage and overcome the limitations of all available sources of evidence, including real-world evidence [8], in order to make lifesaving public health decisions. In particular, Frieden [54] stressed that the very high cost of RCTs and the long timelines needed for planning, recruiting patients, conducting the study, and publishing it, are limitations that *"affect the use of randomized controlled trials for urgent health issues, such as infectious disease outbreaks for which public health decisions must be made quickly on the basis of limited and imperfect data."*

Deaton and Cartwright [55] presented the conceptual framework that underlies RCTs and highlighted several limitations. Among them, they have stressed that randomization requires very large samples on both arms of the trial, otherwise, an RCT should not be presumed to be methodologically superior to a corresponding observational study. For example, the randomized controlled trial study of hydroxychloroquine by Dubee et al. [56], was administratively stopped after recruiting 250 patients, with 124 in the treatment group and 123 in the control group. Although a two-fold mortality rate reduction was observed by day 28, the study failed to reach statistical significance, due to the small sample size. Even if statistical significance had been achieved via a stronger mortality rate reduction signal, the small sample size would have still prevented sufficient randomization. Consequently, although the study has gone through the motions of an RCT, it is not methodologically superior to a retrospective observational study. There are several other RCT studies of hydroxychloroquine with similar shortcomings [57].

Furthermore, although a properly conducted RCT has internal validity, in that the inferences are applicable to the specific group of patients that participated in the trial, the external validity of the RCT outcomes needs to be justified conceptually on the basis of prior knowledge, which is either observational, or based on a deeper understanding of the underlying mechanisms of action. Because COVID-19 mortality risk in the absence of early treatment can span three orders of magnitude (from 0.01% to more than 10%) [58–64], depending on age and comorbidities, trials using low-risk patient cohorts are not informative about expected outcomes on the high-risk patient cohorts and vice versa. Likewise, the timing of treatment and the medication dosage/duration of treatment will confound the results of an RCT. In general, better results are expected when treatment is initiated earlier rather than later, and negative results can be caused by inappropriate medication dosage (i.e., too much or too little). These are all relevant considerations for establishing the external validity of an RCT.

As was noted by Risch [65], when interpreting evidence from RCTs, and more broadly from any study, we should bear in mind that results of efficacy or toxicity of a treatment regimen on hospitalized patients cannot be extrapolated to outpatients and vice versa. Likewise, Risch [65] noted that evidence of efficacy or lack of efficacy of a single drug do

not necessarily extrapolate to using several drugs in combination. This latter point is further amplified when there is an algorithmic overlay governing which drugs should be used and when, based on the individual patient's medical history and ongoing response to treatment. Consequently, RCTs that compare a single drug monotherapy against supportive care are not always informative about whether the drug should be included in a multidrug protocol.

In addition to all that, we are also confronted with an ethical concern. If the available observational evidence are sufficiently convincing, then there is a crossover point where it is no longer ethical to justify randomly refusing treatment to a large number of patients in order to have a sufficiently large control group. The corresponding mathematical challenge is being able to quantify the quality of our observational evidence in order to determine whether or not we are already situated beyond this ethical crossover point.

Just as the quality of evidence provided by randomized controlled trials is fluid, with respect to successful randomization and external validity, the same is true about the quality of real-world evidence [8] that will inevitably become available from the initial response to an emerging new pandemic. We envision that a successful pandemic response, in the area of early outpatient treatment, will proceed as follows: the first element of pandemic response is to assess and monitor the situation by prospectively collecting data, needed to construct predictive models of the probability of hospitalization and death, in the absence of treatments that have yet to be discovered, as a function of the patient's medical profile/history. These models do not necessarily need to be sophisticated at the early stages of pandemic response. It could be sufficient to be able to predict good lower bounds for the hospitalization or mortality probabilities, as opposed to more precise estimates. These early data can be used to identify the predictive factors for hospitalization or death and risk stratify the patients into low-risk and high-risk categories. They can also be used as a historical control group that establishes our prior knowledge of expected outcomes, in the absence of treatment, that has yet to be discovered.

In parallel with gathering and analyzing data, which is the primary duty and responsibility of our public health and academic institutions, medical doctors have an ethical responsibility to use the emerging scientific understanding of the new disease and its mechanisms of actions to try to save the lives of as many patients as possible. Under article 37 of the 2013 Helsinki declaration [66], it is ethically appropriate for physicians to *"use an unproven intervention, if in the physician's judgment it offers hope of saving life, re-establishing health or alleviating suffering"*, provided, there is informed consent from the patient, and *"where proven interventions do not exist, or other known interventions have been ineffective."*

When this effort leads to the discovery of a treatment protocol, with an empirical signal of benefit and acceptable safety, and using the treatment protocol results in a case series of treated patients, then the confluence of the following conditions makes it possible to statistically establish the existence of treatment efficacy: First, the proposed treatment protocols should use repurposed drugs [17] with known acceptable safety. When testing new drugs, we have no prior knowledge of the risks involved and a rigorous controlled study is required to determine the balance of risks and benefits. Second, we need data that give us prior knowledge of the probability risk of the relevant binary endpoints (i.e., hospitalization and/or death) in the absence of treatment, as a function of the relevant stratification parameters. Third, and most importantly, the case series corresponding to treated patients should exhibit a *very strong* signal of benefit, relative to our prior experience with untreated patients, prior to the discovery of the respective treatment protocol.

Under these conditions, the idea that is proposed in this paper works as follows. Our input is the number $N$ of high-risk patients treated, the number of patients $a$ with an adverse outcome (i.e., hospitalization or death) and selection criteria for extracting the high-risk cohort under consideration, from which we can deduce, based on prior knowledge, that the unknown probability $x$ of an adverse outcome without treatment is bounded by $p_1 \leq x \leq p_2$. We also choose the desired level of $p$-value upper bound $p_0$, which is typically $p_0 = 0.05$ (95% confidence), although we shall also consider $p_0 = 0.01$ and $p_0 = 0.001$. The output is an efficacy threshold $x_0(N, a, p_0)$ that gives us the following

rigorous mathematical statement: *if $x_0(N, a, p_0) < x$, then we have more than $1 − p_0$ confidence that the treatment is effective relative to the standard of care.* This statement has to be paired with the subjective assessment of our prior knowledge, based on which we need to show that $x_0(N, a, p_0) < p_1$. The upper bound $p_2$ is used by the Bayesian factor technique as part of finalizing the calculation of the efficacy threshold $x_0(N, a, p_0)$. Implicit in this argument is the assumption of no selection bias, allowing us to apply the probability $x$ at the population level to our particular case series. From the sample size $N$ and the finalized efficacy threshold $x_0$, we also calculate a random selection bias threshold $x_1(N, x_0, p_0)$, higher than $x_0$, that quantifies how large the gap between $p_1$ and $x_0$ needs to be, in order to mitigate with $1 − p_0$ confidence, any possible random selection bias in the case-series sample $(N, a)$.

As a result, we can assert the existence of treatment efficacy using two distinct standards of evidence. If we can establish $x_0 < p_1$, then the *preponderance of evidence* is in favor of the existence of treatment efficacy, and this can justify its provisional adoption on an emergency basis, in order to gather more evidence. If we can establish that $x_1 < p_1$, then the evidence becomes *clear and convincing*, and if these results are replicated by multiple treatment centers, then it becomes ethically questionable to deny patients access to the treatment protocol, for the purpose of conducting an RCT, or simply due to therapeutic nihilism by public health authorities. In Figure 1, we show how the proposed statistical methodology can be integrated into an epidemic or pandemic response that leverages and deploys the direct experience of frontline medical doctors, resulting from their efforts to treat their patients. We stress again that this approach is appropriate only for treatment protocols using repurposed medications with known acceptable safety. When new medications, as opposed to repurposed drugs, are introduced into a pre-existing treatment protocol, then they should be rigorously tested both for safety and efficacy with prospective RCTs.

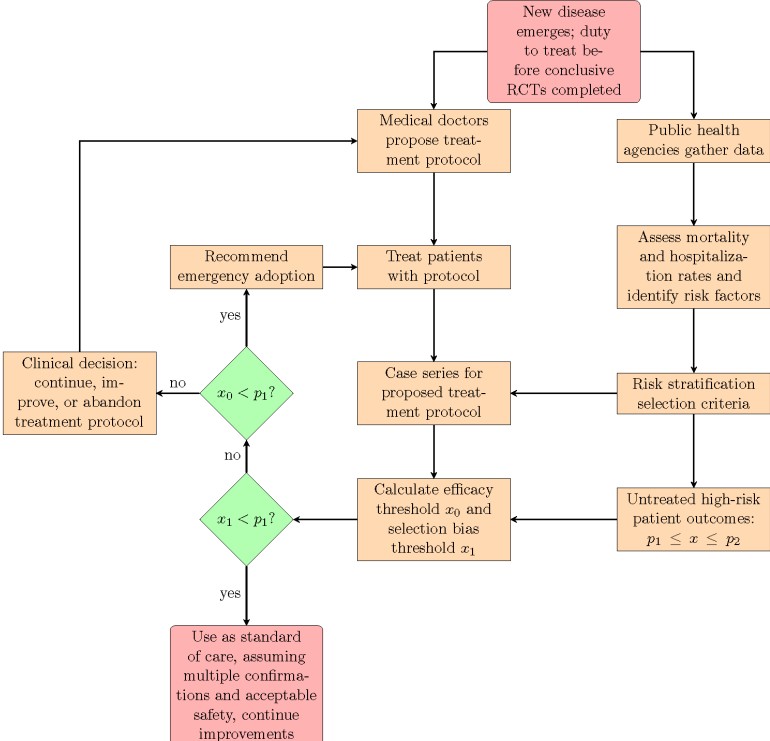

**Figure 1.** This flowchart shows the suggested interactions between medical doctors, public health agencies, and the proposed statistical methodology that are needed, in order to implement an emergency epidemic or pandemic response that leverages the direct experience of frontline medical doctors treating their patients.

The paper is organized as follows. In Section 2 we present the technique for calculating the efficacy threshold and the random selection bias threshold. We also explain the relationship of the proposed technique with the exact Fisher test and with the binomial proportion confidence interval problem. In Section 3, we present a Bayesian technique for adjusting the efficacy thresholds in order to also control the corresponding false positive rate. In Section 4, we illustrate an application of both techniques to the Zelenko case series [2,9,10] as well as the Procter [11,12] and Raoult [13] case series. Discussion and conclusions are given in Section 5. With the exception of Section 3, which is mainly relevant to a more careful analysis by biostatisticians, we have strived to make Sections 2 and 4 of the paper relevant and accessible to both clinicians and biostatisticians by minimizing the mathematical details. Material that is relevant only to biostatisticians is relegated to the appendices. The computer code and the corresponding calculations are included in the supplementary data document [27].

## 2. Methods—Part I: Frequentist Methods for Case Series Analysis

In this section, we present the technique for comparing a treatment group case series of high-risk patients against the expected probability $x$ of an adverse outcome without treatment, based on prior knowledge. Since our prior knowledge bounds the probability $x$ inside an interval $p_1 < x < p_2$ but the precise value of $x$ is unknown, we calculate the minimum value (efficacy threshold) that this probability has to exceed in order to be able to reject the null hypothesis, that the treatment has no efficacy. The proposed technique is equivalent to an exact Fisher test where we take the limit of an infinitely large control group with probability of an adverse outcome set equal to $x$. We also explain the relationship between the proposed approach and the binomial proportion confidence interval problem, and provide evidence that the corresponding coverage probability is conservative. The assumption of conservative coverage is used, in turn, to derive a random selection bias threshold that $x$ should exceed in order reject the possibility of a false positive result due to random selection bias.

### 2.1. Comparing Treatment Group against Expected Adverse Event Rate without Treatment

Suppose that we have a treatment group of high-risk patients in which $N$ patients have received treatment and $a$ patients have had an adverse outcome. Let us also assume that all $N$ patients in the case series satisfy precise selection criteria, used to classify them as high-risk patients, from which we can infer, from our prior knowledge, that in the absence of treatment, the probability $x$ of an adverse outcome for a similar population is bounded in the interval $p_1 \leq x \leq p_2$. To establish the existence of treatment efficacy, we assume the null hypothesis, that the treatment has no effect and that consequently, the probability of an adverse outcome in the treatment group is also equal to $x$. Under this null hypothesis the probability of observing $a$ patients with an adverse outcome out of a total of $N$ patients is given by

$$\mathrm{pr}(N, a|x) = \binom{N}{a} x^a (1 - x)^{N-a}, \tag{1}$$

which corresponds to a binomial distribution. The first factor gives the number of combinations for choosing the $a$ patients that have an adverse outcome out of all $N$ patients. The second factor $x^a$ is the probability that the chosen $a$ patients have an adverse outcome, under the assumption of the null hypothesis. The third factor $(1 - x)^{N-a}$ is likewise the probability that the remaining $N - a$ patients will not have an adverse outcome. Consequently, the product of the three factors is the probability of seeing the event $(N, a)$ under the null hypothesis.

The corresponding $p$-value is calculated by adding to the probability of the event $(N, a)$, the probability of all other events with smaller or equal probability, and it reads

$$p(N, a, x) = \sum_{n=0}^{N} \mathrm{pr}(N, n|x) H(\mathrm{pr}(N, a|x) - \mathrm{pr}(N, n|x)), \tag{2}$$

where $H$ is the modified Heaviside function given by

$$H(x) = \begin{cases} 1, & \text{if } x \geq 0 \\ 0, & \text{if } x < 0 \end{cases}. \tag{3}$$

The Heaviside function factor in Equation (2) selects the events $(N, n)$ that are less probable than the observed event $(N, a)$ for inclusion in the probability sum, as per the formal definition of the *p*-value.

In order to reject the null hypothesis, we need to construct a convincing argument that establishes that $p(N, a, x) < p_0$, with $p_0 = 0.05$ in order to achieve a 95% confidence. Such an argument, in effect, is a hypothesis test that compares the treatment group $(N, a)$ outcome against a fixed probability $x$ for an adverse outcome in the absence of treatment. Our proposal for doing so is conceptually very simple. First we calculate an efficacy threshold $x_0(N, a, p_0)$ such that

$$x_0(N, a, p_0) < x \leq 1 \Longrightarrow p(N, a, x) < p_0. \tag{4}$$

In doing so, we are seeking the smallest possible value of $x_0$ that satisfies Equation (4). If our prior belief about $x$ is that it satisfies $p_1 \leq x \leq p_2$, then it follows that if we show that our prior belief about the lower bound $p_1$ of the probability $x$ of an adverse outcome without treatment exceeds the efficacy threshold $x_0$, then we have a statistically significant signal of benefit in favor of the proposed treatment protocol. This is, in turn, sufficient to recommend to other physicians to consider using the treatment protocol, on an emergency basis, in order to save as many patients as possible, as soon as possible.

We stress again that implicit in this reasoning is the assumption that all observed adverse events in the treatment group case series have been caused by the disease and not by the treatment. For this reason, this methodology has to be limited only to the evaluation of treatment protocols using repurposed medications [17] with previously known acceptable safety. Furthermore, in order to have a prior belief constraining the probability $x$ of an adverse outcome for high-risk patients, in the absence of treatment, it is necessary for public health agencies and academic institutions to prospectively collect data on the predictive factors for hospitalizations or death, as soon as possible, at the beginning of an emerging new disease. These data can then be used both to define the selection criteria for identifying patients as high-risk and to constrain the corresponding probability $x$ within the interval $p_1 \leq x \leq p_2$. Finally, the above argument is predicated on the assumption of no selection bias in the case series $(N, a)$, in order for the inequality $p_1 \leq x \leq p_2$ to be applicable to the case series. We extend the argument to account for selection bias in Section 2.3.

### 2.2. Comments on the Proposed Hypothesis Testing Technique

We now make the following comments about the above hypothesis testing technique. First, we note that the *p*-value $p(N, a, x)$, corresponding to a comparison of a case series $(N, a)$ of a treatment group against the probability $x$ of an adverse outcome without treatment, as given by Equation (2), can be also obtained by running an exact Fisher test with an artificial control group $(M, b)$ of $M$ patients with $b$ adverse outcomes, with $x = b/M$, in the limit where the size of this artificial control group goes to infinity. In Appendix A, we give a mathematical proof of this claim and also explain the mathematically precise formulation of the statement. This convergence property is in fact, a consequence of a known relationship [67–69] between the hypergeometric distribution, used in the calculation of the exact Fisher test *p*-value, and the binomial distribution used in the calculation of $p(N, a, x)$.

Paradoxically, as shown from the example in Figure 2, the convergence of the *p*-value is not monotonic with respect to the control group size. Intuitively, increasing the size of the control group should increase confidence in rejecting the null hypothesis, which should result in a monotonically decreasing *p*-value. Instead, we see that the *p*-value increases,

as the control group size is increased, with intermittent downward jumps driving the convergence to $p(N, a, x)$. We also see that the convergence is slower than we might expect. Nevertheless, the result of Appendix A assures us that, in the limit of an infinite control group, the $p$-value eventually does converge to $p(N, a, x)$.

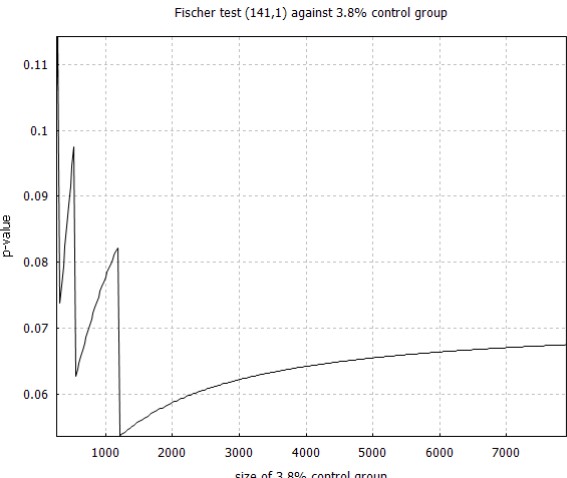

**Figure 2.** We plot the $p$-value calculated from an exact Fisher test that compares the treatment group from the DSZ study [2] (141 high-risk patients treated with 1 death) against an artificial control group with 3.8% mortality rate. Note that the exact $p$-value in the infinite control group limit should be 0.047, which is approached to three decimals when we get to control group size between 160,000 and 180,000.

Second, the proposed hypothesis technique is also mathematically related to the well-researched binomial proportion confidence interval problem [70]. Given the case series $(N, a)$ of a treatment group of $N$ patients, with $a$ patients having an adverse outcome, the challenge of the binomial proportion confidence interval problem is to identify a probability interval $(q_1, q_2)$, such that we can assert, with $1 - p_0$ confidence, that the probability of an adverse event with treatment is inside the interval $(q_1, q_2)$.

If the null hypothesis is satisfied, then the probability of an adverse outcome with treatment is equal to the probability of an adverse event without treatment, and it follows that the intervals $(p_1, p_2)$ and $(q_1, q_2)$ have to intersect. The contrapositive of this deduction is that if the intervals $(p_1, p_2)$ and $(q_1, q_2)$ do not intersect, then the null hypothesis is false. This argument shows that the upper endpoint $q_2$ is the efficacy threshold $x_0(N, a, p_0)$ that has to be exceeded by all probabilities in the interval $(p_1, p_2)$ in order to reject the null hypothesis and claim a signal of benefit. More specifically, the method proposed in the preceding section for calculating the efficacy threshold $x_0(N, a, p_0)$ is equivalent to calculating the upper endpoint of the Sterne interval [71] for the corresponding binomial proportion confidence interval problem.

It is worth noting that although several alternative techniques have been proposed for solving the binomial proportion confidence interval problem, none of them has coverage consistent with the desired statistical confidence and most of them do not have conservative coverage [72]. This means that, given a case series $(N, a)$ for a treatment group, the obtained 95% confidence interval $(q_1, q_2)$ for the probability of an adverse event, with treatment, could be wider or narrower than it should be, depending on the sample size $N$ and the unknown true value of that probability. Furthermore, it has already been proven that no solution to the binomial proportion confidence interval problem exists with perfect coverage [73]. For our purposes, a solution technique with conservative coverage that always overestimates the efficacy threshold $x_0(N, a, p_0)$ is acceptable, and to be preferred over techniques that will sometimes overestimate and sometimes underestimate the efficacy threshold.

The coverage of a specific solution technique is quantified via the coverage probability $c(N, p_0|x)$, which is defined as the conditional probability of observing a case series $(N, a)$,

given a fixed sample size $N$, for which our solution technique will yield a confidence interval that includes the true probability of an adverse event, under the condition that this true probability is equal to $x$. Here, $1 - p_0$ is the desired level of confidence. For very large sample sizes, $c(N, p_0|x)$ is calculated using computer simulations, however for smaller samples, it can be calculated analytically [74] from the equation,

$$c(N, p_0|x) = \sum_{n=0}^{N} I(N, n, x, p_0) \, \text{pr}(N, n|x), \tag{5}$$

where $I(N, n, x, p_0)$ is an indicator function, such that $I(N, n, x, p_0) = 1$ if and only if $x$ is in the confidence interval obtained by the proposed solution technique for a given case series $(N, n)$ corresponding to $1 - p_0$ confidence. Otherwise we set $I(N, n, x, p_0) = 0$. Conservative coverage requires that our solution technique satisfy $c(N, p_0|x) \geq 1 - p_0$ for all values $0 \leq x \leq 1$ of the probability $x$.

The Clopper–Pearson interval [75] is a very well-known solution technique to the binomial proportion confidence interval problem that is known to have conservative coverage. However, an efficacy threshold, defined as the upper limit of the Clopper–Pearson interval, is not equal to what we would have obtained from an exact Fisher test in the limit of an infinite control group, unlike with the Sterne interval [71]. In Figure 3, we show the coverage probability for the Sterne interval for sample sizes $N = 20$ and $N = 100$ and note that it also has conservative coverage, which is very desirable in the context of hypothesis testing. In Figure 4, we compare the coverage probability of the Clopper–Pearson interval against the coverage probability of the Sterne interval and note that although they are both conservative, the Sterne interval has less conservative coverage probability than the Clopper–Pearson interval, over the same sample size.

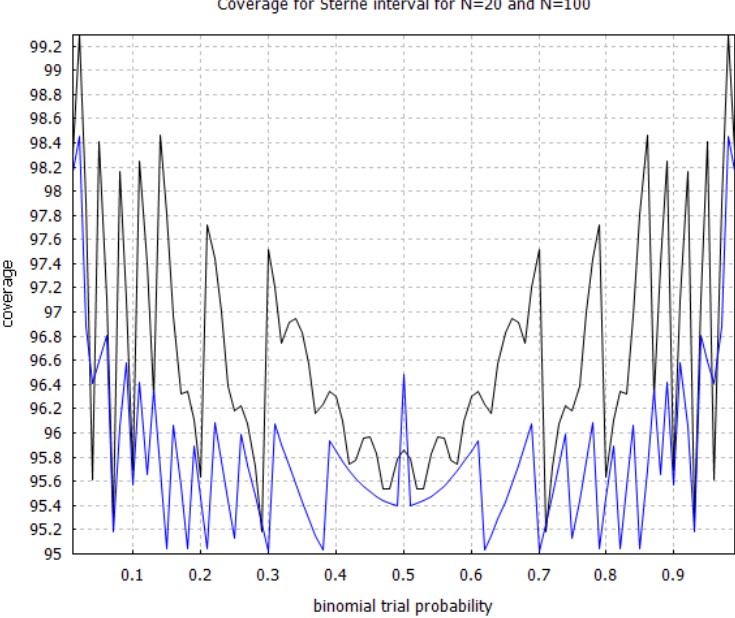

**Figure 3.** Coverage probability for the Sterne interval [71] with sample sizes $N = 20$ and $N = 100$. The black curve corresponds to $N = 20$ and the blue curve, which is situated below the black curve, corresponds to $N = 100$. The coverage probabilities were calculated using 0.01 increments on the horizontal axis.

Our third comment concerns the numerical calculation of the efficacy thresholds $x_0(N, a, p_0)$ from the function $p(N, a, x)$. To illustrate this calculation with an example, on Figure 5, we plot the $p$-value $p(N, a, x)$ against the expected mortality rate $x$ without early outpatient treatment of COVID-19, based on Procter's combined case series [12] of

869 high-risk patients that received an early treatment protocol with 2 reported deaths. The figure has vertical lines marking the crossover to 95%, 99%, and 99.9% confidence. The corresponding efficacy thresholds are located at the points where the zigzag graph of the function $p(N, a, x)$ intersects with the vertical lines. Finding the intersection points numerically with an efficient algorithm is challenging due to the zigzag shape of the graph. An efficient such algorithm was discovered very recently [76], although we did not use it in our calculations [27].

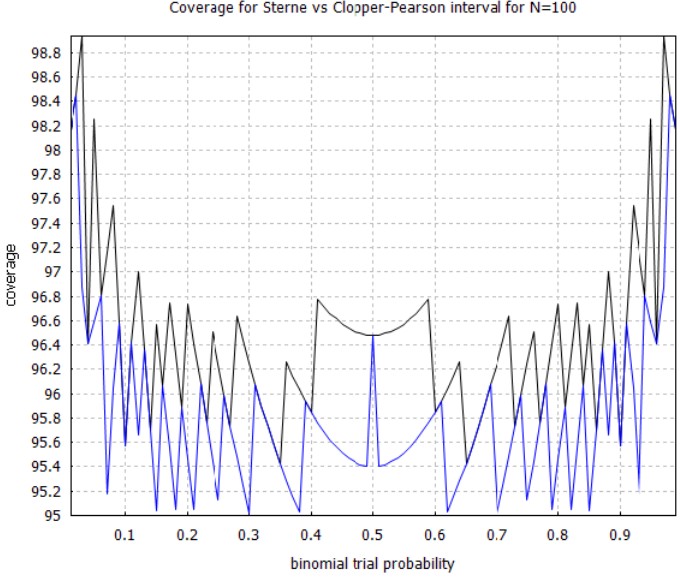

**Figure 4.** Comparison of the coverage probability for the Clopper–Pearson interval [75] versus the Sterne interval [71] with sample size $N = 100$. The black curve shows the coverage probability for the Clopper–Pearson interval, and the blue curve, which is situated below the black curve, shows the coverage probability for the Sterne interval. The coverage probabilities were calculated using 0.01 increments on the horizontal axis.

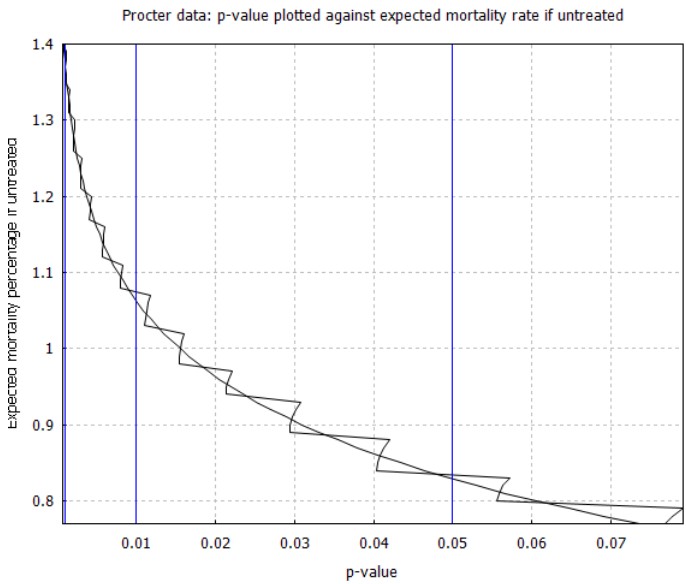

**Figure 5.** Relationship between $p$-value and expected mortality rate for high-risk patients without early treatment, based on the case series data from Procter's dataset of 869 high-risk patients [12]. The zigzag curve follows $p(N, a, x)$ given by Equation (2), whereas the smooth curve approximates the right tail terms in the $p$-value sum by replacing them with the left-tail terms on the horizontal axis.

The discontinuous behavior of $p(N, a, x)$ may seem paradoxical, since we would have expected it to be monotonically decreasing with respect to $x$, but we have found that it is caused by some of the right-tail contributions to the $p$-value sum given by Equation (2). If we make an unwarranted approximation, replacing the right-tail sum with the left-tail sum, we obtain the smooth curve shown in Figure 5. The intersection points of the smooth curve with the vertical lines, give the upper endpoint of the Clopper–Pearson interval [75]. Similar graphs for all case series considered in the study have been included in our supplementary data document [27]. We have found empirically, at least for the case series being studied here, that both the correct zigzag curve and the approximate smooth curve give almost the same values for all relevant efficacy thresholds.

*2.3. Selection Bias Mitigation and Selection Bias Thresholds*

The idea of hypothesis testing, in which we compare a case series of treated patients against the historical population level (or possibly a more limited) control group, is vulnerable to the criticism of possible selection bias in the treatment case series in favor of establishing treatment efficacy. Some of the selection bias could be systemic (i.e., there may be a tendency towards selecting healthier high-risk patients), but even in the absence of any systemic bias, some selection bias will inevitably occur randomly, as a consequence of using a small sample of patients for the treatment case series, randomly chosen out of the general population. We propose the following idea for mitigating random selection bias, and then we discuss the problem of selection bias more broadly.

Suppose that we have a case series $(N, a)$, of $N$ treated patients with $a$ adverse outcomes, and suppose that we have calculated the efficacy threshold $x_0$ for this case series. We can choose $x_0$ to be either set equal to $x_0(N, a, p_0)$, or we can choose to have it further increased, if necessary, using the Bayesian technique of Section 3. In either case, if we have a prior belief that the probability of an adverse event, without treatment, in the high-risk part of the general population, under the same high-risk patient selection criteria used to form the case series, is equal to $x$, then, when selecting a random sample of $N$ high-risk patients out of the general population, we can have $1 - p_0$ confidence that the true rate $x'$ of adverse events, without treatment, for that particular sample, will range according to a discretized confidence interval $m_1(N, x, p_0)/N \leq x' \leq m_2(N, x, p_0)/N$. Here, $m_1(N, x, p_0)$ is the minimum number of adverse events and $m_2(N, x, p_0)$ is the maximum number of adverse events that we expect to see in any one particular sample of $N$ high-risk patients, in the absence of treatment, with confidence $1 - p_0$. The possible criticism of our approach is that, perhaps, for the specific sample of patients in our case series, the true adverse event rate $x'$, without treatment, could happen to be below the efficacy threshold $x_0$, in spite of the corresponding population level adverse event rate $x$ exceeding the efficacy threshold $x_0$. This raises the question of how big does the gap between $x$ and $x_0$ need to be, to ensure that the entire confidence interval of $x'$ lies above the efficacy threshold $x_0$? Since the lower endpoint of this confidence interval is $m_1(N, x, p_0)/N$, the answer to this question defines a new higher threshold $x_1(N, x_0, p_0)$, which we shall call the *random selection bias threshold*, if set is equal to the minimum value of $x_1$ that satisfies

$$x_1(N, x_0, p_0) < x \leq 1 \implies x_0 < m_1(N, x, p_0)/N. \tag{6}$$

In Appendix B, we prove that an upper bound of this random selection bias threshold can be calculated by choosing the smallest possible value of $x_1$ that satisfies the implication

$$x_1(N, x_0, p_0) < x \leq 1 \implies p(N, \lceil x_0 N \rceil, x) < p_0. \tag{7}$$

Here, the notation $\lceil x_0 N \rceil$ represents rounding the number $x_0 N$ upwards towards the nearest integer. We note that the calculation of the confidence interval for $x'$, as given by Appendix B, uses the assumption that the Sterne interval solution [71] of the binomial proportion confidence interval problem has conservative coverage. Given the random selection bias threshold $x_1$, and the prior knowledge that in a similarly high-risk cohort, at

the population level, the probability $x$ of an adverse outcome, in the absence of treatment, ranges between $p_1 \leq x \leq p_2$, establishing that $x_1 < p_1$ with some gap between $x_1$ and $p_1$ can be used to rule out random selection bias, with $1 - p_0$ confidence, as the sole cause of a signal of efficacy.

Generally speaking, it is more likely than not that a strong efficacy signal cannot be caused in a particular observation, solely as a result of random selection bias, as long as $x$, which is near the center of the confidence interval for $x'$, exceeds the efficacy threshold $x_0$. Even if part of the $x'$ confidence interval is below $x_0$, more than half of the interval will be above $x_0$. As a result, the efficacy threshold $x_0$ and the random selection bias threshold $x_1$ quantify two levels of evidence. Showing $x_0 < p_1$ establishes the existence of treatment efficacy by the *preponderance of evidence*. Meeting this evidentiary standard should be sufficient for communicating the proposed treatment protocol to other physicians for emergency adoption, with a caveat that it is still investigational, and that more data are needed before making a definitive claim. Showing $x_1 < p_1$ establishes the existence of treatment efficacy by the *clear and convincing* evidentiary standard. Our view is that exceeding the random selection bias threshold $x_1$, for a treatment protocol with acceptable safety, is an objective milestone beyond which therapeutic nihilism, and even the denial of treatment for research purposes, becomes unethical.

With regards to the broader problem of systemic bias, there are multiple possibilities to consider: there may be some population-level geographic bias in the patients that live in the geographic area served by a particular treatment center; there may be reporting bias, in that we hear about case series because of the good outcomes, without these outcomes being representative of the actual outcomes at the national or international level; there may be bias in the patient demographics (ratio of low vs. high-risk patients), and with respect to the timing of treatment (early vs. late treatment). The latter concern can be addressed by stratifying the case series with respect to risk and/or timing of treatment. Geographic bias can be addressed by investigating case series across multiple geographic locations and/or by using localized population statistics for the historical control. Outcome reporting bias can be minimized, if we have consecutive case series from the same treatment center, where the initially reported outcomes are replicated by subsequent results.

Further mitigation of systemic selection bias is possible by establishing a large gap between the random selection bias threshold $x_1$ and the lower bound $p_1$ for adverse outcomes in the historical control statistics. To quantify the magnitude of systemic selection bias, consider the likelihood ratio $L = x/(1-x)$ of selecting unhealthy vs. healthy patients, if the selection is truly random, i.e., without any systemic bias. Here, we define *unhealthy patients* to be the high-risk patients that will have an adverse outcome without early treatment, if symptomatically infected, and we define *healthy patients* to be the patients that are not unhealthy patients. If there is systemic bias in favor of selecting healthy patients, then that could account for a false positive signal of efficacy. It would also reduce the corresponding likelihood ratio to $L/F$, with $F \geq 1$ a numerical factor measuring how much more likely it is to choose healthy patients due to systemic selection bias. In Appendix B, we have also shown that the systemic selection bias threshold $x_1(F|N, x_0, p_0)$, that $x$ has to overcome in order to mitigate systemic selection bias with magnitude $F$, is related to the random selection bias threshold $x_1(N, x_0, p_0)$ via the equation:

$$x_1(F|N, x_0, p_0) = \frac{F x_1(N, x_0, p_0)}{1 + (F-1)x_1(N, x_0, p_0)}. \tag{8}$$

Given our prior belief that, at the population level, the probability $x$ of an adverse outcome in high-risk patients without treatment satisfies $p_1 \leq x \leq p_2$, we can find the maximum amount $F_{\max}$ of selection bias that can be tolerated, before the evidence quality

falls back to the *preponderance of evidence* evidentiary standard, by solving the equation $x_1(F|N, x_0, p_0) = p_1$ with respect to $F$. The corresponding solution is given by

$$F_{\max} = \frac{p_1[1 - x_1(N, x_0, p_0)]}{x_1(N, x_0, p_0)(1 - p_1)}. \tag{9}$$

This means that if the systemic bias tends to select healthy high-risk patients $F$ times more likely than the likelihood corresponding to their proportion in the general population of high-risk patients, then $1 \leq F < F_{\max}$ implies that we can have at least $1 - p_0$ confidence that the observed positive signal of efficacy cannot be explained solely as a consequence of systemic selection bias.

Last, but not least, statistical quantitative evidence can be corroborated and amplified with more qualitative evidence based on the Bradford Hill criteria [77] for establishing a causal relationship between treatment and positive patient outcomes. Particularly relevant are the criteria of: (1) *plausibility*, i.e., the existence of a known biological mechanism of action that explains why the treatment protocol is expected to work; (2) *consistency*, i.e., observing the same effect in different treatment centers in different locations; (3) *biological gradient*, i.e., observing improved outcomes with increased medication dosage or length of treatment, additional medications, or by initiating treatment earlier, rather than later; (4) *temporality*, i.e., immediate improvement in symptoms, following the administration of the treatment protocol. The statistical evidence alone speak in support of only the *strength of association* criterion, but that is only one of the several criteria proposed by Bradford Hill [77]. If we can establish that these additional criteria are satisfied, then that constitutes additional evidence on top of the statistical evidence that treatment efficacy exists and that the signal of benefit cannot be explained away, as a result solely caused by selection bias.

### 3. Methods—Part II: Bayesian Factor Analysis of Efficacy Thresholds

The methodology that we proposed in Section 2 is also vulnerable to the criticism that rejecting the null hypothesis, solely on the basis that the $p$-value satisfies $p < 0.05$ is not sufficient for asserting that treatment efficacy is statistically significant. This is indeed the position of the recent American Statistical Association statement on statistical significance and $p$-values [19]. The problem is that $p$-values only measure how incompatible the data are with the null hypothesis. Consequently, a concern has been expressed that it is not self-evident that the $p$-value will always do a good job at controlling the probability of a false positive result [78]. To estimate the latter probability, we would have to formulate the appropriate alternate hypothesis and consider how much the data are compatible or incompatible with that alternate hypothesis. This has prompted recommendations to lower the $p$-value threshold down to 0.01 or 0.001 [78,79]. However, this is only a stopgap measure that does not fundamentally address the problem.

In this section, we supplement the $p$-value based analysis of Section 2 with a proposal for a Bayesian factor analysis [20–24]. The Bayesian factor compares the alternate hypothesis (treatment efficacy) against the null hypothesis, and can be used to calculate the probability of a false positive result [25]. We do not mean to suggest that the Bayesian factor should replace the $p$-value in hypothesis testing. Our view is that we need to use both. That is, use the $p$-value to reject the null hypothesis, and then use the Bayesian factor to assess the strength of the evidence in favor of the alternate hypothesis. This viewpoint is similar to earlier proposals for conditional frequentist testing [20].

In the following, we will briefly review the Bayesian factor framework, and outline our specific proposal for validating and adjusting, as needed, the efficacy threshold $x_0(N, a, p_0)$. We note that the calculation of the random selection bias threshold $x_1$ is independent of the technique used to calculate the efficacy threshold $x_0$. In terms of procedure, one could initially calculate the efficacy threshold $x_0$ using only the technique of Section 2, and use that to calculate the corresponding random selection bias threshold $x_1$. Alternatively, a more detailed analysis would involve: (a) calculating the efficacy threshold using the technique of Section 2; (b) adjusting the efficacy threshold $x_0$ using the technique presented

in this section; (c) using the adjusted efficacy threshold $x_0$ to calculate the corresponding random selection bias threshold $x_1$.

### 3.1. Bayesian Factor and the False Positive Rate

Let $A, B$, be two arbitrary events in some probability space. From the definition of conditional probability, we obtain the Bayes rule, given by

$$p(B|A) = \frac{p(A|B)p(B)}{p(A)}. \tag{10}$$

Let $D$ represent our data, $H_0$ represent the null hypothesis and $H_1$ represent the alternate hypothesis. In the Bayesian statistics framework, we assign probabilities $p(H_0), p(H_1)$ to the hypotheses $H_0, H_1$ representing our prior belief about how likely each hypothesis is, and then calculate the updated probabilities $p(H_0|D)$ and $p(H_1|D)$ on the condition of observing the data $D$. In this way, Bayesian statistics is distinct from frequentist statistics where probabilities are not assigned to the hypotheses themselves. From the Bayes rule we have,

$$p(H_1|D) = \frac{p(D|H_1)p(H_1)}{p(D)}, \tag{11}$$

$$p(H_0|D) = \frac{p(D|H_0)p(H_0)}{p(D)}, \tag{12}$$

and dividing the two equations gives

$$\frac{p(H_1|D)}{p(H_0|D)} = \frac{p(D|H_1)}{p(D|H_0)} \frac{p(H_1)}{p(H_0)}. \tag{13}$$

The Bayes factor $B(D|H_1, H_0)$ is defined to read

$$B(D|H_1, H_0) = \frac{p(D|H_1)}{p(D|H_0)}, \tag{14}$$

and it is the numerical factor that amplifies our prior belief about the odds ratio $b(H_1, H_0) = p(H_1)/p(H_0)$ after seeing the data $D$. Here, $p(D|H_1)$ is the probability of seeing the data $D$ if $H_1$ is true and $p(D|H_0)$ is likewise the probability of seeing the data $D$ if $H_0$ is true.

To interpret the meaning of the Bayesian factor, the following argument is used to calculate the posterior probabilities $p(H_1|D)$ and $p(H_0|D)$ in terms of $B(D|H_1, H_0)$ and $b(H_1, H_0) = p(H_1)/p(H_0)$. We assume that $H_0, H_1$ satisfy $p(H_0) + p(H_1) = 1$ and $p(H_0|D) + p(H_1|D) = 1$. Combining the second equation with Equations (11) and (12) gives the Bayes theorem

$$p(D) = p(D|H_0)p(H_0) + p(D|H_1)p(H_1), \tag{15}$$

and it follows that the probability of a false positive result is given by

$$p(H_0|D) = \frac{p(D|H_0)p(H_0)}{p(D)} \tag{16}$$

$$= \frac{p(D|H_0)p(H_0)}{p(D|H_0)p(H_0) + p(D|H_1)p(H_1)} \tag{17}$$

$$= \frac{p(D|H_0)p(H_0)}{p(D|H_0)p(H_0)[1 + B(D|H_1, H_0)b(H_1, H_0)]} \tag{18}$$

$$= \frac{1}{1 + B(D|H_1, H_0)b(H_1, H_0)}. \tag{19}$$

We see that the false positive probability $p(H_0|D)$ approximately scales as the inverse of the Bayes factor $B(D|H_1, H_0)$. On the other hand, the dependence of $p(H_0|D)$ on the prior likelihood ratio $b(H_1, H_0)$, which measures our subjective belief about the odds ratio between $H_1$ and $H_0$, before seeing the data $D$, is uncomfortable. There are three ways to cope with that: First, one can simply join the frequentist camp, consider probabilities based on beliefs as meaningless, and forget about the whole thing. Second, one can use an uninformed prior, meaning that we assume that both hypotheses $H_0$ and $H_1$ are equally probable, not having any prior knowledge that favors one over the other, and choose $p(H_0) = p(H_1) = 1/2$, which corresponds to $b(H_1, H_0) = 1$. An interesting third way is to use the reverse Bayesian analysis technique proposed by Colquhoun [25], which is based on the equivalence

$$p(H_0|D) < p_0 \iff b(H_1, H_0) > \frac{1 - p_0}{p_0 B(D|H_1, H_0)}, \tag{20}$$

which relates an upper bound $p_0$ on the probability $p(H_0|D)$ with a corresponding lower bound $b_{\min}(p_0, B)$ on the prior likelihood ratio $b(H_1, H_0)$, which is given by

$$b_{\min}(p_0, B) = \frac{1 - p_0}{p_0 B}, \tag{21}$$

with $B$ being the value of the corresponding Bayesian factor. The meaning of Equation (21) is that, given a desired lower bound $p_0$ for the false positive rate and a threshold $B$ for the Bayesian coefficient, $b_{\min}(p_0, B)$ is the minimum prior likelihood ratio $p(H_1)/p(H_0)$ for our prior knowledge of the extent to which the alternate hypothesis $H_1$ is favored over the null hypothesis $H_0$, for which the Bayesian threshold $B$ can control the false positive rate and keep it below $p_0$. As such, given our subjective choice for $b_{\min}$, one can calculate the threshold $B$ for the Bayesian factor corresponding to the minimum tolerated false positive rate $p_0$.

Since we wish to constrain the false positive rates to less than 0.05, in order to claim 95% statistical significance, we choose $p_0 = 0.05$. Kass and Raftery [24] and Jeffries [80] both recommend that the threshold $B > 100$ be used for a *decisive* acceptance of the alternate hypothesis $H_1$ over the null hypothesis $H_0$. Using $B = 100$, we find that $b_{\min}(0.05, 100) = 0.19$. This means that if we associate the decisive threshold $B > 100$ with 95% confidence, doing so is equivalent to a prior belief that the null hypothesis is 5 times more likely than the alternate hypothesis. In turn, this prior belief can be used to deduce Bayesian factor thresholds for higher levels of confidence, consistently with our choice to associate $B > 100$ with 95% confidence. This choice can be interpreted as defining the word "decisive" to mean 95% confidence, in the context of stating that $B > 100$ is "decisive". It could be critiqued as being an arbitrary choice, but the same can be said for the $p_0 = 0.05$ p-value threshold and the Bayesian factor $B > 100$ threshold. Our particular approach has the advantage of being more transparent, in terms of an intuitive interpretation, than an arbitrary choice made in terms of the prior probabilities for $H_0$ and $H_1$.

### 3.2. Application to Hypothesis Testing for Case Series

Now, let us consider how Bayesian factor analysis can be applied to a case series with a treatment group of $N$ patients, where $a$ patients have an adverse outcome. Let $x_0$ be the corresponding efficacy threshold, determined via the techniques of Section 2, and let $x$ be the probability of an adverse outcome *with* treatment. We define a null hypothesis $H_0$ and an alternate hypothesis $H_1$ about the value of $x$ such that

$$H_0 : x_0 < x \leq 1, \tag{22}$$
$$H_1 : 0 < x \leq x_0. \tag{23}$$

We use for $x_0$ the upper endpoint of the binomial proportion confidence interval corresponding to the observed data $(N, a)$. Consequently, the null hypothesis $H_0$ has been defined to place $x$ outside and above that interval, and the alternative hypothesis $H_1$ considers the remaining possible values for $x$.

Because both $H_0$ and $H_1$ are composite hypotheses, it is necessary to introduce prior probabilities $\mathrm{pr}(x|H_0)$ and $\mathrm{pr}(x|H_1)$, corresponding to $H_0$ and $H_1$. It may seem tempting to just use uninformed priors for both $H_0$ and $H_1$, however, doing so would certainly not be appropriate for the null hypothesis $H_0$ in almost all situations, since with many illnesses, we can rule out the probability of an adverse outcome exceeding some upper bound $p_2$. Instead, we can thus use an uninformed prior on the interval $[x_0, p_2]$, given by

$$\mathrm{pr}(x|H_0(x_0, p_2)) = \begin{cases} 1/(p_2 - x_0), & \text{if } x \in [x_0, p_2] \\ 0, & \text{if } x \in (p_2, 1], \end{cases} \tag{24}$$

and perform an appropriate sensitivity analysis on the parameter $p_2$. In general, increasing $p_2$ will tend to increase the Bayes factor, since doing so will tend to increase the contrast between the null and alternate hypotheses. So, we can explore how much $p_2$ can be decreased and still maintain a decisive Bayes factor. Likewise, for the alternate hypothesis $H_1$, we will use an uninformed prior on the interval $[0, t]$ with $t \leq x_0$ given by

$$\mathrm{pr}(x|H_1(x_0, t)) = \begin{cases} 1/t, & \text{if } x \in [0, t] \\ 0, & \text{if } x \in (t, x_0]. \end{cases} \tag{25}$$

The reason for this choice is that we have found empirically that, in some cases, the Bayes factor may actually increase if, instead of an uninformed prior on $[0, x_0]$, we use an uninformed prior on the shorter interval $[0, t]$. From an intuitive standpoint, we surmise that if the data has a very strong efficacy signal, then the contrast between the null and alternate hypotheses is increased when one eliminates the relatively unlikely values of $x$ between $t$ and $x_0$. For this reason, we shall use the maximum value of the Bayes factor taken over all values $t \in (0, x_0)$, on a decimal logarithmic scale which is given by

$$b(x_0, p_2) = \max_{t \in (0, x_0]} b_0(x_0, p_2, t), \tag{26}$$

$$b_0(x_0, p_2, t) = \log B(N, a|H_1(x_0, t), H_0(x_0, p_2)). \tag{27}$$

In Appendix C we prove that the function $b_0(x_0, p_2, t)$ is initially increasing and then decreasing, with respect to $t$, with a maximum in the interval $[a/N, 1]$. If this maximum is located in the narrower interval $[a/N, x_0]$, then the optimal Bayes factor is indeed obtained when we use a choice $t \in (0, p_0)$ for the prior distribution of the alternate hypothesis $H_1$. If the maximum is formally located at $t > x_0$, then the optimal Bayes factor is obtained at $t = x_0$. The resulting metric $b(x_0, p_2)$ is still dependent on the parameter $p_2$ of the prior distribution of the null hypothesis $H_0$.

To complete the metric definition by Equations (26) and (27), we now show the calculation of the Bayes factor $B(N, a|H_1(x_0, t), H_0(x_0, p_2))$ between $H_1$ and $H_0$ as a function of $x_0, p_2, t$ and the data $N, a$. We note that the probabilities for seeing the data $(N, a)$, under the hypotheses $H_1$ and $H_0$, are given by:

$$\mathrm{pr}(N, a|H_0(x_0, p_2)) = \int_{x_0}^{1} \mathrm{d}x \, \mathrm{pr}(N, a|x) \, \mathrm{pr}(x|H_0(x_0, p_2)) \tag{28}$$

$$= \frac{1}{p_2 - x_0} \int_{x_0}^{p_2} \mathrm{d}x \, \mathrm{pr}(N, a|x) \tag{29}$$

$$= \frac{1}{p_2 - x_0} \binom{N}{a} \int_{x_0}^{p_2} x^a (1 - x)^{N-a} \, \mathrm{d}x, \tag{30}$$

and

$$\mathrm{pr}(N, a | H_1(x_0, p_2)) = \int_0^{x_0} \mathrm{d}x \, \mathrm{pr}(N, a | x) \, \mathrm{pr}(x | H_1(x_0, t)) \tag{31}$$

$$= \frac{1}{t} \int_0^t \mathrm{d}x \, \mathrm{pr}(N, a | x) \tag{32}$$

$$= \frac{1}{t} \binom{N}{a} \int_0^t x^a (1 - x)^{N-a} \, \mathrm{d}x, \tag{33}$$

consequently, the corresponding Bayes factor is given by

$$B(N, a | H_1(x_0, t), H_0(x_0, p_2)) = \frac{\mathrm{pr}(N, a | H_1(x_0, p_2))}{\mathrm{pr}(N, a | H_0(x_0, p_2))} \tag{34}$$

$$= \frac{p_2 - x_0}{t} \frac{\displaystyle\int_0^t x^a (1 - x)^{N-a} \, \mathrm{d}x}{\displaystyle\int_{x_0}^{p_2} x^a (1 - x)^{N-a} \, \mathrm{d}x}. \tag{35}$$

The integrals can be calculated using exact algebra or numerically with the open source computer algebra software Maxima [81]. The exact algebra calculation takes longer to carry out, but we have confirmed that the numerical calculation using the function quad_qagr is just as accurate.

In order to control for the false positive rate, we propose that the efficacy thresholds $x_0(N, a, p_0)$ with $p_0 = 0.05$ should be increased, if necessary, by requiring that they also satisfy $b(x_0, p_2) \geq 2$. Since the threshold used for a decisive Bayes factor with $p_0 = 0.05$ corresponds approximately to $b_{\min}(p_0, B) = 1/5$, it is reasonable to use the empirical formula

$$b(x_0, p_2) \geq \log\left(\frac{5(1 - p_0)}{p_0}\right), \tag{36}$$

to adjust the efficacy thresholds $x_0(N, a, p_0)$ for an arbitrary value of demanded confidence $p_0$. For $p_0 = 0.01$, Equation (36) gives $b(x_0, p_2) \geq 2.7$. For $p_0 = 0.001$, we find $b(x_0, p_2) \geq 3.7$. Both Bayes factor thresholds also correspond to a prior likelihood ratio $p(H_1)/p(H_0) = 1/5$. Consequently, they are the thresholds that we recommend imposing on the Bayes factors for the purpose of adjusting the corresponding efficacy thresholds $x_0(N, a, p_0)$ for the choices $p_0 = 0.01$ and $p_0 = 0.001$.

## 4. Results

We shall now apply the proposed framework to the processing of available high-risk COVID-19 patient case series by Zelenko [2,9,10], Procter [11,12], and Raoult [13] that provide evidence for the original Zelenko triple-drug protocol [2] and the more advanced McCullough protocol [14–16], which are both based on safe repurposed medications. Section 4.1 reviews the case series under consideration. Section 4.2 summarizes the data and the calculation of the corresponding efficacy and random selection bias thresholds. These are used in Sections 4.3 and 4.4 to assess the evidence in support of mortality rate reduction and hospitalization rate reduction correspondingly. Section 4.5 shows that the Bayesian factor analysis of the efficacy threshold has negligible impact for the specific case series under consideration.

### 4.1. Review of the Zelenko, Procter and Raoult Case Series

In the Zelenko April 2020 letter [9], Zelenko reported on his outcomes based on a total of 1450 patients that he treated for COVID-19 until 28 April 2020 in an Orthodox Jewish community in upstate New York. From this cohort, 405 patients were classified as high risk and treated with his triple-drug therapy (hydroxychloroquine, azithromycin, zinc sulfate). The reported outcomes were six hospitalizations and two deaths. From amongst the patients classified as low risk, who were given only supportive care, there were no

hospitalizations or deaths. Zelenko's criteria for risk stratification define three categories of high-risk patients: (1) every patient older than 60; (2) every patient younger than 60 but with comorbidities or obesity (BMI (Body Mass Index) $\geq 30 \text{kg/m}^2$); (3) patients younger than 60 and without comorbidities that presented with shortness of breath.

A subset of the 28 April 2020 case series was published in a case controlled study [2] that included only the patients seen with COVID-19 infection that was confirmed by a PCR (Polymerase Chain Reaction) test or an antibody IgG (Immunoglobulin G) test. The remaining patients were clinically diagnosed from symptomatic presentation and via ruling out a bacterial or influenza infection. This Derwand–Scholz–Zelenko study (hereafter DSZ study) [2] included 335 patients of which 141 patients were classified as high-risk patients and treated with the triple-drug protocol, with 4 hospitalizations and 1 death. Detailed demographic data are given for the high-risk patient treatment group, including a detailed breakdown in the three high-risk categories. The study also included a control group of 377 patients, who were seen by other treatment centers in the same community, that were only offered supportive care and no early outpatient treatment. From this untreated group, 13 patients died and 58 patients were hospitalized. The untreated group includes both low-risk and high-risk patients, so we expect that it underestimates both the hospitalization and mortality risk for high-risk patients. Unfortunately, demographic data were not available for the untreated group, so from a strictly methodological point of view, one cannot entirely rule out the theoretical possibility that the untreated group might have consisted of patients that are at higher risk, on average, than those of the high-risk treatment group. On the other hand, using a case series of untreated patients from Israel [61], with demographic data indicating a combination of low and high-risk patients, with 143 deaths reported out of 4179 untreated patients, gives the same 3.4% mortality rate as in the DSZ control group, suggesting that the DSZ control group also consists of a mixed demographic of low- and high-risk patients.

The June 2020 Zelenko case series [10] is reported in a letter that Zelenko sent to the Israeli Health Minister at the time, Dr. Moshe Bar Siman-Tov, on 14 June 2020, which was later made publicly available. In the letter, Zelenko reported that a total of approximately 2200 patients were seen as of 14 June 2020, with 800 patients deemed high-risk, under the same criteria who were treated with the triple-drug therapy, since the beginning of the pandemic. The reported cumulative outcomes are: 12 hospitalizations, 2 deaths, no serious side effects, and no cardiac arrhythmias.

During the April 2020–June 2020 interval, at the beginning of May 2020, Zelenko enhanced his triple-drug therapy protocol with oral dexamethasone and budesonide nebulizer. He introduced the blood thinner Eliquis towards the end of May 2020 and the beginning of June 2020. Ivermectin was not used by Zelenko until October 2020. Consequently, the DSZ study [2] and the Zelenko April 2020 case series [9] reflect the outcomes of the triple-drug therapy, when used by itself as an early outpatient treatment. The Zelenko June 2020 case series [10] includes the use of steroid medications and a blood thinner, so the underlying treatment protocol is closer to the McCullough protocol [14–16].

It is worth noting that both letters [9,10] were originally posted on Google Drive by Zelenko, and were censored by Google during 2021. The April 2020 letter [9] was cited by Risch [65], whose paper has also preserved the corresponding case series data. The June 2020 letter case series data [10] was independently reported in a subsequent publication by Risch [82], which included only the number of reported deaths, and not the number of hospitalizations. The authors have attached copies of all three Zelenko letters [9,10,26] to our supplementary material document [27].

The Procter case series were reported consecutively in two publications [11,12]. The first paper [11] reports on 922 patients that were seen between April 2020 and September 2020, of which 320 were risk stratified as high-risk patients and treated with the McCullough protocol [14–16]. The outcome was six hospitalizations and one death. The second paper [12] reports on an additional patient cohort seen between September 2020 and December 2020. Out of the total number of patients during that time period, 549 were risk stratified



as high-risk and treated with an outcome of 14 hospitalizations and one death. For both case series, the risk stratification criteria were similar to those used by Zelenko. However, the age threshold used to risk stratify patients as high-risk was lowered to 50 years. The medications used were customized for each patient in accordance with the McCullough protocol [14–16] and included hydroxychloroquine, ivermectin, zinc, azithromycin, doxycycline, budesonide, foliate, thiamin, IV fluids, and for more severe cases, dexamethasone and ceftriaxone were also added. Demographic details for the cohorts were reported in the respective publications [11,12].

The final high-risk patient case series is extracted from a recent cohort study [13] of 10,429 patients that were seen between March 2020 and December 2020 by Raoult's IHU Méditerranée Infection hospital in Marseille, France. From the entire cohort, 8315 patients were treated with hydroxychloroquine, azithromycin, and zinc. Of those patients, those older than 70 or with comorbidities were also treated with enoxaparin. Low-dose dexamethasone was given on a case by case basis to patients that presented with inflammatory pneumonopathy, high viral loads, or on a case by case basis. This treatment protocol is consistent, to some extent, with the principles that underlie the McCullough protocol [14–16]. The remaining 2114 patients did not receive hydroxychloroquine or azithromycin or both due to contraindications or because the patients did not consent to using one or two of these medications. This cohort was used in the Raoult study [13] as a control group. The study risk-stratified the patients by age (see Table 1 of Ref. [13]), making it possible to extract a case series of high-risk patients under the restriction age $\geq 60$. In the treatment group, this results in 1495 high-risk patients with 5 deaths and 106 hospitalizations. In the control group, under the age $\geq 60$ constraint, there are 520 high-risk patients with 38 hospitalizations and 11 deaths. The authors note that no serious adverse events to the medications were reported, and that the reported deaths were not related to side effects of hydroxychloroquine or azithromycin. Furthermore, no deaths were reported for age $< 60$ cohort in both the treatment group and control group.

### 4.2. Tabular Summaries of the Zelenko, Procter, and Raoult Case Series

Table 1 summarizes the aforementioned case series, including the treatment groups from the DSZ study [2], the Zelenko [2,9,10] and Procter [11,12] case series, and the age $\geq 60$ treatment group from the Raoult study [13]. Note that the Zelenko June 2020 case series and the Procter II case series as reported on Table 1, combine the two respective consecutive case series. We also report in Table 1 the DSZ study's control group [2], the alternative Israeli control group [61], and the age $\geq 60$ part of the Raoult control group [13]. We emphasize that all reported treatment group case series consist of high-risk patients.

From a cursory examination of Table 1, we see that the mortality rate is consistent across all treatment groups, which speaks to the consistency Bradford Hill criterion [77]. Hospitalization rates are also consistent between the Zelenko [2,9,10] and Procter case series [11,12], but there is a clear discrepancy with the hospitalization rates reported in the Raoult treatment case series [13]. We believe that the reason for the discrepancy is that both Zelenko and Procter explicitly aimed to prevent hospitalizations due to the poor outcomes of the inpatient treatment protocols used in the United States. In Marseille, France, Raoult had the option of using his IHU Méditerranée Infection hospital for short hospitalizations, in order to closely monitor his more concerning cases.

In Table 2, we show the results of comparing the Zelenko April 2020 [9] and Zelenko June 2020 [10] case series against both the original DSZ control group [2] as well as the alternative control group from Israel [61]. The confidence intervals were calculated using Woolf's formula [83,84]. Although in the original DSZ study [2] mortality rate reduction was not statistically significant, we have found that comparing either the Zelenko April 2020 case series [9] or the June 2020 case series [10] against either control group, gives more than 90% mortality rate reduction, which is also statistically significant in terms of both *p*-value and confidence interval. Likewise, we see at least 90% hospitalization rate reduction when the Zelenko April 2020 case series or Zelenko June 2020 case series is

compared against the DSZ control group, which is statistically significant as well. Because the control groups consist of a combination of both low-risk and high-risk patients, whereas the treatment groups consist of only high-risk patients, the resulting comparisons are biased towards the null, and thus underestimate the actual efficacy of the respective treatment protocols. This comparison is compelling due to the consistency between the two control groups, as evidence in favor of showing the existence of treatment efficacy. However, from a methodological standpoint, it may not be convincing enough by itself in terms of measuring the extent of treatment efficacy.

**Table 1.** Case series list: The table lists the total number of patients, the subset of high-risk patients that were treated with a sequenced multidrug regimen, number of patients that were hospitalized, and number of deaths, for the following case series: Derwand–Scholtz–Zelenko study treatment group [2], Zelenko's complete April 2020 data set [9], Zelenko's complete June 2020 data set [10], Procter's observational studies [11,12], and Raoult's high-risk (older than 60) treatment group [13]. The table also lists the same data for the control group in the DSZ study [2], the untreated group in the Israeli study [61], and the control group in the Raoult study [13].

| Study | Total | High-Risk | Hospitalizations | Deaths |
|---|---|---|---|---|
| Case series data from Refs. [2,9–13] | | | | |
| DSZ study [2] | 712 | 141 | 4 (2.8%) | 1 (0.7%) |
| Zelenko 04/2020 [9] | 1450 | 405 | 6 (1.4%) | 2 (0.4%) |
| Zelenko 06/2020 [10] | 2200 | 800 | 12 (1.5%) | 2 (0.25%) |
| Procter I [11] | 922 | 320 | 6 (1.8%) | 1 (0.3%) |
| Procter II [12] | ? | 869 | 20 (2.3%) | 2 (0.2%) |
| Raoult [13] | 10429 | 1495 | 106 (7.0%) | 5 (0.3%) |
| Control group data from Refs. [2,13,61] | | | | |
| DSZ control [2] | 377 | <377 | 58 (>15%) | 13 (>3.4%) |
| Israeli control [61] | 4179 | <4179 | N/A | 143 (>3.4%) |
| Raoult control [13] | 2114 | 520 | 38 (7.3%) | 11 (2%) |

**Table 2.** Exact Fisher test comparing the mortality rate reduction and hospitalization rate reduction between the high-risk patient treated group the DSZ study [2], Zelenko's complete April 2020 data set [9], and Zelenko's complete June 2020 data set [10] against the low risk and high-risk patient control groups in the DSZ study [2] and the Israeli study [61]. The *p*-values where there is a failure to establish 95% confidence are highlighted with bold font.

| Study | Odds Ratio | 95% CI | *p*-Value |
|---|---|---|---|
| Exact Fisher tests on mortality rates | | | |
| DSZ study vs. DSZ control | 0.2 | 0.02–1.54 | **0.12** |
| Zelenko 04/2020 vs. DSZ control | 0.13 | 0.03–0.61 | 0.003 |
| Zelenko 06/2020 vs. DSZ control | 0.07 | 0.01–0.31 | $10^{-5}$ |
| DSZ vs. Israeli control | 0.2 | 0.03–1.45 | **0.09** |
| Zelenko 04/2020 vs. Israeli control | 0.14 | 0.03–0.57 | 0.0002 |
| Zelenko 06/2020 vs. Israeli control | 0.07 | 0.02–0.28 | $10^{-9}$ |
| Exact Fisher tests on hospitalization rates | | | |
| DSZ vs. DSZ control | 0.16 | 0.05–0.45 | 0.02 |
| Zelenko 04/2020 vs. DSZ control | 0.08 | 0.03–0.19 | $10^{-13}$ |
| Zelenko 06/2020 vs. DSZ control | 0.08 | 0.04–0.16 | $10^{-19}$ |

We have also calculated the efficacy threshold for mortality rate reduction and hospitalization rate reduction corresponding to the case series by Zelenko [2,9,10], Procter [11,12], and Raoult [13]. The calculations are shown in the supplementary material document [27]. The results are tabulated in Table 3. We display the efficacy thresholds for 95%, 99%, and 99.9% confidence, which are calculated as the upper end points of the corresponding Sterne

interval [71] and, in parentheses, we display the corresponding random selection bias thresholds. We use precision of 0.1% for most case series, except for the two largest ones, Procter II [12] and Raoult [13], where we use 0.01% precision.

Each threshold corresponds to a mathematically rigorous conditional statement about rejecting the null hypothesis that the corresponding early outpatient treatment protocol is ineffective. For example, the 1.8% efficacy threshold corresponding to 95% confidence for rejecting the null hypothesis in the Zelenko April 2020 case series [9] corresponds to the following statement: *If the expected mortality rate for an equivalent cohort without early outpatient treatment exceeds 1.8%, then the null hypothesis can be rejected with at least 95% confidence.* Similar statements can be formulated for each efficacy threshold metric on Table 3. Likewise, the 4.0% random selection bias threshold for the Zelenko April 2020 case series [9] corresponds to the following statement: *If the observed mortality rate, at the population level, for high-risk patients, classified as such using the same selection criteria as in the treatment case series, exceeds 4.0%, then we can be 95% confident that the observed signal of efficacy cannot be attributed solely to random selection bias, and we can also reject the null hypothesis with at least 95% confidence.* Similar statements are implied from all of the other random selection bias thresholds reported on Table 3.

**Table 3.** Mortality and hospitalization rate reduction efficacy thresholds, defined as the upper end of the Sterne interval [71], corresponding to 95%, 99%, and 99.9% confidence, for the DSZ study treatment group [2], Zelenko's complete April 2020 data set [9], Zelenko's complete June 2020 data set [10], Procter's observational studies [11,12], and Raoult's high-risk (older than 60) treatment group [13]. In parenthesis, we also display the corresponding higher random selection bias thresholds.

| Study | 95% Threshold | 99% Threshold | 99.9% Threshold |
|---|---|---|---|
| Mortality rate efficacy thresholds | | | |
| DSZ study | 3.8% (9.2%) | 5.3% (12.8%) | 7.0% (14.6%) |
| Zelenko 04/2020 | 1.8% (4.0%) | 2.4% (5.2%) | 2.9% (6.9%) |
| Zelenko 06/2020 | 1.0% (2.0%) | 1.2% (2.7%) | 1.6% (3.7%) |
| Procter I | 1.7% (4.1%) | 2.3% (5.8%) | 3.1% (7.8%) |
| Procter II | 0.84% (1.82%) | 1.08% (2.46%) | 1.4% (3.37%) |
| Raoult | 0.79% (1.40%) | 0.96% (1.87%) | 1.18% (2.46%) |
| Hospitalization rate efficacy thresholds | | | |
| DSZ study | 7.0% (12.7%) | 8.8% (17.5%) | 10.6% (21.5%) |
| Zelenko 04/2020 | 3.2% (5.4%) | 3.9% (7.2%) | 4.7% (9.5%) |
| Zelenko 06/2020 | 2.7% (4.2%) | 3.0% (5.0%) | 3.5% (6.4%) |
| Procter I | 4.1% (7.3%) | 4.9% (9.1%) | 5.9% (11.6%) |
| Procter II | 3.6% (5.2%) | 4.0% (6.1%) | 4.5% (7.5%) |

These statements are mathematical facts. However, to complete the inference argument, they need to be paired with an inevitably subjective statement that provides an estimate, or at least a lower bound, on the expected mortality or hospitalization rates of similar cohorts without early outpatient treatment. Secondarily, we need an inference about the intervals of mortality or hospitalization rates, in the absence of early outpatient treatment, in order to do the Bayesian adjustment of the efficacy thresholds.

*4.3. Analysis of Mortality Rate Reduction Efficacy*

To establish that early treatment protocols result in mortality rate reduction, when administered to high-risk patients, we recall that patients have been classified as high-risk based on the following three categories: (1) old age; (2) comorbidities or obesity (with BMI $\geq 30$ kg/m$^2$); (3) shortness of breath upon presentation. The age threshold for high risk classification is age $\geq 60$ for the Zelenko [2,9,10] and Raoult [13] case series, and age $\geq 50$ for the Procter [11,12] case series. The high-risk treatment groups for the Zelenko [2,9,10] and Procter [11,12] case series include the demographic distribution of

all three categories of high-risk patients, whereas in the Raoult [13] case series we have included only age $\geq$ 60 patients. Our approach, in the following, is to lower bound the mortality rate, in the absence of early outpatient treatment, separately for each of the three high-risk patient categories. Then, the common lower bound becomes applicable to any demographic distribution of the three categories. To establish the existence of treatment efficacy, it is sufficient for this lower bound to exceed the corresponding thresholds of Table 3. In the following, we shall now consider the mortality rate for each of the three high-risk patient categories separately.

With regards to the first category of patients classified as high-risk due to old age, the earliest data from China [60], as of 11 February 2020, estimated a minimum of 3.6% mortality rate for patients older than 60 and a minimum of 1.3% mortality rate for patients older than 50 (see Table 4). These numbers are consistent with numbers from China [58] and Italy [59] as of March 17, 2020 (see Table 5). We can also estimate the mortality risk of the first category of high-risk patients (age $\geq$ 60 or age $\geq$ 50) using adjusted estimates by the CDC (Centers for Disease Control and Prevention) [62–64] of COVID-19 deaths per symptomatic cases. The CDC report attempts to adjust for the differences in under-reporting of symptomatic illness, hospitalizations, and deaths, and it is based on reports ranging from February 2020 to September 2021. The raw data and a copy of the CDC report website are given in our supplementary material document [27]. From that, we calculate for the age $\geq$ 50 group a mortality rate of 2.26% (95% CI: 1.94– 2.61%). We cannot deduce an age $\geq$ 60 mortality rate from the CDC report, but note that the age $\geq$ 65 mortality rate, according to the CDC, is 4.79% (95% CI: 4.11% to 5.52%). We observe that the stratification of mortality risk with respect to age is consistent between three distinct geographical regions.

**Table 4.** Crude case fatality rate data, in the absence of early outpatient treatment, based on early data from China as of 11 February 2020, and published on 30 March 2020. [60]. We highlight with bold font the high-risk age brackets with CFR $\geq$ 1.0%.

| Age | Deaths | Cases | CFR |
|---|---|---|---|
| 10–19 | 0 | 416 | 0% |
| 20–29 | 7 | 3619 | 0.193% |
| 30–39 | 18 | 7600 | 0.237% |
| 40–49 | 38 | 8571 | 0.4% |
| **50–59** | 130 | 10,008 | **1.3%** |
| **60–69** | 309 | 8583 | **3.6%** |
| **70–79** | 312 | 3918 | **7.96%** |
| **$\geq$80** | 208 | 1408 | **14.8%** |
| **$\geq$60** | 829 | 13,909 | **5.96%** |

**Table 5.** Crude case fatality rate data, in the absence of early outpatient treatment, based on early data from China and Italy as of 17 March 2020 and published on 23 March 2020 [58,59]. We highlight with bold font the high-risk age brackets with CFR $\geq$ 1.0%.

| Age | Italy CFR | China CFR |
|---|---|---|
| 0–9 | 0% | 0% |
| 10–19 | 0% | 0.2% |
| 20–29 | 0% | 0.2% |
| 30–39 | 0.3% | 0.2% |
| 40–49 | 0.4% | 0.4% |
| **50–59** | **1.0%** | **1.3%** |
| **60–69** | **3.5%** | **3.6%** |
| **70–79** | **12.8%** | **8.0%** |
| **$\geq$80** | **20.2%** | **14.8%** |

The second category of high-risk patients are patients with comorbidities regardless of age. In Table 6, we show case fatality rates with respect to comorbidities (i.e., cardiovascular disease, diabetes, respiratory disease, hypertension, cancer), based on data from China [58] in the period up to 11 February 2020, and additional data from Israel [61], with patients diagnosed in the period up to 16 April 2020 and deaths recorded up to July 16, 2020. There is variability in mortality rates from 5% to 15%. The Israeli data appear to show higher mortality rates than the data from China, and the reason for that could be that the Israeli study [61] accounted for the time lag between patient diagnosis and death. Nevertheless, with respect to using 5% as a lower bound mortality rate for high-risk patients with comorbidities, the available data from both locations are consistent.

These studies do not account for the mortality risk from obesity, and do not account for the mortality risk corresponding to the third category of high-risk patients that present with shortness of breath. A collaborative study by Risch and a research group in Brazil [85] found, using multivariate regression analysis, that both obesity and dyspnea pose a higher mortality risk than heart disease (see Table 2 of Ref. [85]), therefore, we expect that they both lie in the same 5% to 15% interval as patients with other comorbidities.

**Table 6.** Case fatality rate based on early-stage analysis of COVID-19 outbreak in China in the period up to 11 February 2020 [58] vs. similar statistics from Israel published on 7 September 2020 [61].

| Comorbidity | Deaths | Cases | CFR |
|---|---|---|---|
| Comorbidity CFR from Chinese study [58] | | | |
| Cardiovascular disease | 92 | 873 | 10.5% |
| Diabetes | 80 | 1102 | 7.3% |
| Respiratory disease | 32 | 511 | 6.3% |
| Hypertension | 161 | 2683 | 6% |
| Cancer | 6 | 107 | 5.6% |
| Comorbidity CFR from Israeli study [61] | | | |
| Cardiovascular disease | 87 | 518 | 16.7% |
| Diabetes | 71 | 531 | 13% |
| Respiratory disease | 23 | 361 | 6% |
| Hypertension | 102 | 744 | 13.7% |
| Cancer | 37 | 264 | 10% |

For the case of obesity, as a mortality risk factor, this conclusion is also supported by more recent meta-analysis [86], showing that obesity is a greater mortality risk factor than diabetes and hypertension, and one that increases with increasing BMI. A study of 148,494 patients across 238 hospitals by the CDC [87] also confirms that obesity is an increasing mortality risk factor with increasing BMI. It is known that obesity is associated with increased levels of the inflammatory cytokines TNF-$\alpha$ (tumor necrosis factor alpha), IL-1$\beta$ (interleukin-1-beta), and IL-6 (interleukin 6), produced by macrophages in the adipose tissue [88]. A study of 9390 hospitalized patients in Abu Dhabi, United Arab Emirates, has found that patients with severe COVID-19 symptoms, requiring intensive care, had significantly elevated IL-6 biomarker relative to patients that presented with mild or moderate symptoms [89]. An earlier meta-analysis [90] has also confirmed that the IL-6 biomarker is associated with severe progression of the COVID-19 disease. Consequently, there is a very compelling biological mechanism that explains why obesity is a severe risk factor for progression of the disease to the COVID-19 pneumonia phase, requiring a high risk classification and immediate early outpatient treatment.

For the case of patients presenting with shortness of breath, it is important to appreciate the fact that, without an early outpatient treatment intervention, such presentation implies that the disease is progressing beyond the viral replication phase, into the COVID-19 pneumonia phase, soon to be followed with the thromboembolic stage, oxygen desaturation,

and hospitalization. It is thus self-evident that these patients should be classified as high-risk and treated immediately. Assuming that most of such patients will be hospitalized without outpatient treatment, we can also estimate the corresponding mortality risk, in the absence of outpatient treatment, by looking at the conditional probability of death, assuming hospitalization has already taken place. A study by the Houston Methodist Hospital [91] has shown an average mortality rate of 5.8% for hospitalized patients between March 2020 and July 2020, in spite of the use of hydroxychloroquine and anticoagulants. Furthermore, the study reports 12.1% mortality rate, for hospitalized patients between 13 March 2020 and 15 May 2020, and 3.5% mortality rate between 16 May 2020 and 7 July 2020, corresponding to two consecutive surges, noting that the second surge targeted younger patients than the first surge. A prospective multicenter study [92] from Italy of 1050 patients in the Coracle registry, between 22 February 2020 and 1 April 2020, showed an overall 13% average mortality rate, with 7.4% mortality rate for hospitalized patients that do not require supplemental oxygen or invasive ventilation, 12.8% mortality rate for hospitalized patients that require supplemental oxygen, and 22.9% mortality rate for hospitalized patients that are invasively ventilated.

Based on the above arguments, we can lower bound the untreated mortality risk by 3.5% for the age $\geq$ 60 demographic and by 5.0% for the high-risk patients with comorbidities, obesity, or shortness of breath presentation. For the age $\geq$ 50 demographics, we have an expected 2.26% mortality rate for the United States demographic distribution, as estimated by the CDC. The common lower bound for high-risk patients in all three categories of the Zelenko case series is thus estimated as 3.5%, it exceeds the efficacy thresholds for both the Zelenko April 2020 [9] and the Zelenko June 2020 [10] case series, and it also exceeds the random selection bias threshold for the Zelenko June 2020 case series [10] with $F_{max} \geq 1.77$. The same untreated mortality rate lower bound of 3.5% applies to the Raoult case series [13], which exceeds the efficacy threshold of 0.79% and the random selection bias threshold of 1.40% by a wide margin with $F_{max} \geq 2.55$. Finally, using the CDC mortality rate of 2.26%, which includes a minority of treated patients and a majority of untreated patients for the age $\geq$ 50 demographic in the United States, as a conservative untreated mortality rate lower bound for the Procter case series, we find that it exceeds the efficacy threshold for both Procter I [11] and Procter II [12] case series, and also exceeds the random selection bias threshold for the Procter II case series [12] with $F_{max} \geq 1.24$. We stress that the estimates for $F_{max}$ are lower bounds, and note that the results from the Raoult case series [13] are particularly robust against systemic selection bias.

A completely different approach is to compare the efficacy and random selection bias thresholds against the CFR for the entire population [93]. The CFR for the United States and France is displayed on Figure 6 for the time period between April 2020 and October 2021. During 2020, the CFR ranged from 2% to 6% in the United States and from 2% to 16% in France. In both countries, the CFR converged to 1.7% during 2021 and remained roughly constant, with very small oscillations throughout 2021. The minimum value of 1.7% exceeds the mortality rate reduction efficacy thresholds for the Zelenko June 2020 [10], Procter II [12], and Raoult case series [13]. It also exceeds the random selection bias threshold for the Raoult case series [13]. Using 2.0% as the minimum CFR during 2020, we note that it exceeds the random selection bias threshold for the Procter II case series [12] and it equals the random selection bias threshold for the Zelenko June 2020 case series [10]. Taking the CFR at face value, this is a very strong signal of efficacy, because the CFR includes asymptomatic, low-risk, and high-risk patients, regardless of whether they received early treatment, against solely high-risk patients in the treatment groups of the respective case series. This comparison strongly biases against being able to reject the null hypothesis, but we are still able to do so.

In particular, we note that in the United States, the CFR ranged from 2% to 6% during 2020, which lies above the 1.8% mortality rate reduction efficacy threshold for Zelenko April 2020 case series [9]. This is an indicator that the *preponderance of evidence* was in favor of adopting Zelenko's triple-drug protocol at that time, on an emergency basis, but was

nonetheless not officially adopted in the United States for outpatients [94]. By June 2020, the respective efficacy threshold decreased to 1.0%, and the random selection bias threshold decreased to 2.0%, while the CFR was still in the neighborhood of 3.0%. Thus, the evidence in favor of adopting the Zelenko triple-drug therapy had just crossed over to the *clear and convincing* evidentiary standard by the summer of 2020. Raoult's data [13] were available by December 2020, and strongly corroborate Zelenko's results [9,10,28]. In particular, the $F_{max} \geq 2.55$ lower bound, obtained for the Raoult case series [13], means that even if there is systemic selection bias in favor of selecting healthy high-risk patients by a factor of 2.55, we can be 95% confident that the observed signal of benefit cannot be explained by systemic selection bias alone.

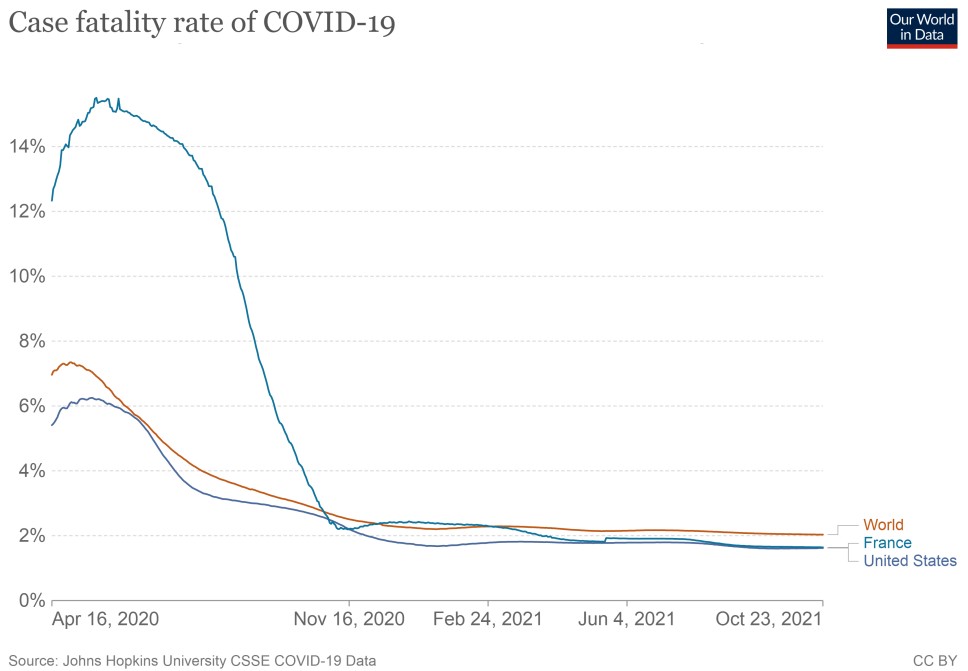

**Figure 6.** Cumulative case fatality rate in the United States and France between April 2020 and November 2021.

### 4.4. Analysis of Hospitalization Rate Reduction Efficacy

In Table 3, we see that the 95% efficacy thresholds for hospitalization rate reduction range from 2.7% to 4.1% for all case series, with the exception of the DSZ case series, where it is at 7.0% due to the smaller sample size. Likewise, the random selection bias thresholds for hospitalization rate reduction with 95% confidence range from 4.2% to 7.3% for all case series, except for the DSZ case series [2].

These thresholds can be compared against the following empirical data. At the beginning of the pandemic, based on data from China until February 11, 2020, there was an initial estimate [60] that the probability of hospitalization for a high-risk age $\geq 60$ cohort would range from 10% to 18%. The control group from Zelenko's study [2], consisting of both low and high-risk patients, again at the beginning of the pandemic here in the United States, reported 377 patients with 58 hospitalizations, corresponding to 15% hospitalization rate. In the Cleveland study [95], which was used to train a predictive model for the risk of hospitalization and death based on patient medical history, the entire data set consisted of a total of 4536 patients between 8 March 2020 and 5 June 2020. There were 582 hospitalizations corresponding to 21% hospitalization rate. In the Mass General Brigham hospital study [96], from a cohort of 12,347 patients that tested positive, there were 3401 hospitalizations between 4 March 2020 and 14 July 2020, corresponding to a 27% hospitalization rate. This was also a cohort that included both low-risk and high-risk patients. The CDC adjusted data [62–64] between February 2020 and September 2021, estimate a 13.79% (95% CI:

17.09% to 28.52%) hospitalization probability for the age $\geq 50$ group, given a symptomatic infection. For the age $\geq 65$ cohort, this estimate increases to 22.09% (95% CI: 17.09% to 28.52%)

Overall, our observation is that we tend to see numbers ranging from 10% to 28% with substantial variability between various cohorts, all of which were not given early outpatient treatment. On the other hand, we see that the case series of high-risk patients shown in Table 3, have efficacy thresholds for hospitalization rate reduction ranging from 2.7% to 4.1%, which have a substantial separation from the 10% to 28% interval. Most remarkably, the hospitalization rate reduction random selection bias thresholds also have a substantial separation from the 10% to 28% interval. We interpret this big gap between the two intervals as strong evidence of the existence of hospitalization rate reduction efficacy as a result of the respective early outpatient treatment protocols in the Zelenko April 2020 [9], Zelenko June 2020 [10], Procter I [11], and Procter II case series [12]

*4.5. Bayesian Analysis Of Efficacy Thresholds*

We shall now assess whether the efficacy thresholds need to be increased, using the Bayesian technique described in Section 3, in order to control the false positive rate. In Table 7, we have calculated the logarithmic Bayesian metric $b(x_0, p_2)$, given by Equation (26), for the mortality and hospitalization rate reduction efficacy thresholds corresponding to 95% confidence, using a range of values of $p_2$ for the purpose of sensitivity analysis. The calculation details are available in our supplementary material document [27]. Recall from Section 3 that $p_2$ corresponds to our sense of the worst possible probability of the respective adverse outcome (hospitalization or death) in high-risk patients, in the absence of early outpatient treatment. As such, 5% to 10% is a typical range for mortality rates in untreated high-risk patients, making $p_2 = 5\%$ a highly conservative choice. We did not consider values higher than 10%, even though worse probabilities are possible, because for $p_2 > 10\%$, we see that all logarithmic Bayesian factors already satisfy $b(x_0, p_2) \geq 2$. We have also looked at $p_2 = 2\%$, which is obviously entirely unrealistic, because it corresponds to the mortality rate of the Raoult control group [13], where some partial treatment was given. Likewise, for the hospitalization rate reduction efficacy thresholds, we have used the values $p_2 = 10\%, 15\%, 20\%$ based on our expectation of a typical 10% to 28% range for the probability of hospitalization, in the absence of early outpatient treatment. We did not consider $p_2 > 20\%$ since almost all of the logarithmic Bayesian factors satisfy $b(x_0, p_2) \geq 2$ at $p_2 = 20\%$.

In Table 8, we compare the efficacy thresholds for rejecting the null hypothesis with the corresponding 95% confidence Bayesian thresholds, obtained by the inequality $b(x_0, p_2) \geq 2$ for accepting the alternate hypothesis. For the DSZ study [2], we see that the corresponding Bayesian thresholds for hospitalization rate reduction range from 7.2% to 9.5%, which lie above the 7.0% threshold obtained via the $p$-value. So, the most cautious course of action is to opt for the 9.5% threshold, which is still below most of our estimates for hospitalization probability of untreated patients. For the DSZ study [2], for both $p_2 = 2\%$ and $p_2 = 5\%$, the logarithmic Bayesian factor for mortality rate reduction does not go above the decisive threshold for any value of $x$ with $a/N \leq x \leq p_2$, consequently the corresponding Bayesian thresholds are undefined, and for $p_2 = 10\%$ we find a Bayesian mortality rate reduction threshold of 3.9% which is slightly larger than the $p$-value threshold of 3.8%. For the Procter I case series [11], there is a weak indication that the 4.1% efficacy threshold for hospitalization rate reduction might have to be increased to 4.3%, and the mortality rate reduction threshold increased from 1.7% to 1.9%. Likewise for the Procter II case series [12], an increase of the hospitalization rate reduction efficacy threshold from 3.6% to 3.7% is weakly indicated. Both adjustments are negligible and inconsequential. For the Zelenko April 2020 [9] and Zelenko June 2020 [10] case series, where the sample sizes are much larger, we see that the overall trend is for the Bayesian thresholds to be far more lenient than the ones obtained via the $p$-value. This is possibly attributed to a very strong signal of efficacy in the data.

**Table 7.** Bayes factor (decimal logarithm) corresponding to the 95% efficacy threshold (Sterne interval [71]) for mortality and hospitalization rate reduction, using maximum untreated mortality rate $p_2$ for high-risk patients at $p_2 \in \{0.02, 0.05, 0.10\}$ and maximum untreated hospitalization rate $p_2$ for high-risk patients at $p_2 \in \{0.10, 0.15, 0.20\}$, for the DSZ study treatment group [2], Zelenko's complete April 2020 data set [9], Zelenko's complete June 2020 data set [10], Procter's observational studies [11,12], and Raoult's high-risk (older than 60) treatment group [13]. We highlight in bold font the Bayes factors that violate the condition $b(x_0, p_2) \geq 2$.

| Bayes factors at the mortality rate efficacy thresholds | | | | |
|---|---|---|---|---|
| **Study** | **95% threshold** | **log Bayes factors** | | |
| | | $p_2 = 0.02$ | $p_2 = 0.05$ | $p_2 = 0.1$ |
| DSZ study | 3.8% | **N/A** | **1.38** | **1.99** |
| Zelenko 04/2020 | 1.8% | **1.17** | 2.04 | 2.45 |
| Zelenko 06/2020 | 1.0% | 2.06 | 2.66 | 3.02 |
| Procter I | 1.7% | **1.28** | 2.07 | 2.47 |
| Procter II | 0.84% | **1.92** | 2.48 | 2.82 |
| Raoult | 0.79% | **1.91** | 2.45 | 2.79 |

| Bayes factors at the hospitalization rate efficacy thresholds | | | | |
|---|---|---|---|---|
| **Study** | **95% threshold** | **log Bayes factors** | | |
| | | $p_2 = 0.10$ | $p_2 = 0.15$ | $p_2 = 0.20$ |
| DSZ study | 7.0% | **1.30** | **1.71** | **1.92** |
| Zelenko 04/2020 | 3.2% | 2.00 | 2.24 | 2.39 |
| Zelenko 06/2020 | 2.7% | 2.24 | 2.47 | 2.61 |
| Procter I | 4.1% | **1.89** | 2.15 | 2.32 |
| Procter II | 3.6% | **1.98** | 2.23 | 2.39 |

It is interesting to repeat the Bayesian analysis on the efficacy thresholds for mortality rate reduction and hospitalization rate reduction corresponding to 99% confidence and 99.9% confidence. We have seen that the Bayesian adjustments to the 95% confidence efficacy thresholds, when they are needed, are very small, so the relevant question is whether this pattern continues when the demanded confidence increases to 99% or 99.9%. Tables 9 and 10 show the values of the logarithmic Bayesian factor $b_2(x_0, p_2)$ at the mortality and hospitalization efficacy thresholds for 99% and 99.9% confidence, as determined solely from the $p$-value, and for various values of $p_2$, as previously discussed. Note that for Table 9, the decisive Bayesian factor threshold corresponding to 99% confidence is $b_2(x_0, p_2) \geq 2.7$. Likewise, in Table 10, the decisive Bayesian factor threshold corresponding to 99.9% confidence is $b_2(x_0, p_2) \geq 3.7$. We see that the logarithmic Bayesian factors are either above or near their respective thresholds.

Likewise, in Tables 11 and 12, we are comparing the mortality and hospitalization rate reduction efficacy thresholds determined via the $p$-value against the corresponding efficacy thresholds determined using the logarithmic Bayesian factor $b_2(x_0, p_2)$ for 99% and 99.9% confidence correspondingly. We see that the Bayesian perturbations to the efficacy thresholds are mostly negligible for both 99% and 99.9% confidence, continuing the similar pattern that we have observed for the 95% confidence efficacy thresholds.

Based on these results, we conclude that for the case series under consideration, the Bayesian adjustments to the efficacy thresholds for mortality and hospitalization rate reduction are negligible, and they do not impact the analysis of the preceding sections.

**Table 8.** Comparison of the 95% confidence efficacy threshold (Sterne interval [71]) for mortality and hospitalization rate reduction with the Bayes factor efficacy thresholds at log Bayes = 2, using maximum untreated mortality rate $p_2$ for high-risk patients at $p_2 \in \{0.02, 0.05, 0.10\}$ and maximum untreated hospitalization rate $p_2$ for high-risk patients at $p_2 \in \{0.10, 0.15, 0.20\}$, for the DSZ study treatment group [2], Zelenko's complete April 2020 data set [9], Zelenko's complete June 2020 data set [10], Procter's observational studies [11,12], and Raoult's high-risk (older than 60) treatment group [13]. We highlight in bold font the Bayesian thresholds that exceed the frequentist thresholds.

| Mortality rate Bayesian efficacy thresholds | | | | |
|---|---|---|---|---|
| **Study** | **95%** | **log Bayes = 2 thresholds** | | |
| | **threshold** | $p_2 = 2\%$ | $p_2 = 5\%$ | $p_2 = 10\%$ |
| DSZ study | 3.8% | **N/A** | **N/A** | **3.9%** |
| Zelenko 04/2020 | 1.8% | **N/A** | 1.8% | 1.5% |
| Zelenko 06/2020 | 1.0% | 1.0% | 0.8% | 0.6% |
| Procter I | 1.7% | **N/A** | **1.9%** | 1.3% |
| Procter II | 0.84% | **0.87%** | 0.7% | 0.6% |
| Raoult | 0.79% | **0.82%** | < 0.7% | < 0.7% |

| Hospitalization rate Bayesian efficacy thresholds | | | | |
|---|---|---|---|---|
| **Study** | **95%** | **log Bayes = 2 thresholds** | | |
| | **threshold** | $p_2 = 10\%$ | $p_2 = 15\%$ | $p_2 = 20\%$ |
| DSZ study | 7.0% | **9.5%** | **7.8%** | **7.2%** |
| Zelenko 04/2020 | 3.2% | **3.2%** | 3.0% | 2.9% |
| Zelenko 06/2020 | 2.7% | 2.6% | 2.5% | 2.4% |
| Procter I | 4.1% | **4.3%** | 4.0% | 3.7% |
| Procter II | 3.6% | **3.7%** | 3.5% | 3.4% |

**Table 9.** Bayes factor (decimal logarithm) corresponding to the 99% efficacy threshold (Sterne interval [71]) for mortality and hospitalization rate reduction, using maximum untreated mortality rate $p_2$ for high-risk patients at $p_2 \in \{0.02, 0.05, 0.10\}$ and maximum untreated hospitalization rate $p_2$ for high-risk patients at $p_2 \in \{0.10, 0.15, 0.20\}$, for the DSZ study treatment group [2], Zelenko's complete April 2020 data set [9], Zelenko's complete June 2020 data set [10], Procter's observational studies [11,12], and Raoult's high-risk (older than 60) treatment group [13]. We highlight in bold font the Bayes factors that violate the condition $b(x_0, p_2) \geq 2.7$.

| Bayes factors at the mortality rate efficacy thresholds | | | | |
|---|---|---|---|---|
| **Study** | **99% threshold** | **log Bayes factors** | | |
| | | $p_2 = 0.02$ | $p_2 = 0.05$ | $p_2 = 0.1$ |
| DSZ study | 5.3% | **N/A** | **N/A** | 2.70 |
| Zelenko 04/2020 | 2.4% | **N/A** | 2.81 | 3.27 |
| Zelenko 06/2020 | 1.2% | **2.53** | 3.21 | 3.57 |
| Procter I | 2.3% | **N/A** | 2.72 | 3.17 |
| Procter II | 1.08% | **2.55** | 3.17 | 3.53 |
| Raoult | 0.96% | **2.57** | 3.16 | 3.51 |

| Bayes factors at the hospitalization rate efficacy thresholds | | | | |
|---|---|---|---|---|
| **Study** | **99% threshold** | **log Bayes factors** | | |
| | | $p_2 = 0.10$ | $p_2 = 0.15$ | $p_2 = 0.20$ |
| DSZ study | 8.8% | **1.83** | **2.42** | **2.67** |
| Zelenko 04/2020 | 3.9% | 2.75 | 3.00 | 3.17 |
| Zelenko 06/2020 | 3.0% | 2.77 | 3.00 | 3.16 |
| Procter I | 4.9% | **2.55** | 2.85 | 3.02 |
| Procter II | 4.0% | **2.63** | 2.89 | 3.05 |

**Table 10.** Bayes factor (decimal logarithm) corresponding to the 99.9% efficacy threshold (Sterne interval [71]) for mortality and hospitalization rate reduction, using maximum untreated mortality rate $p_2$ for high-risk patients at $p_2 \in \{0.02, 0.05, 0.10\}$ and maximum untreated hospitalization rate $p_2$ for high-risk patients at $p_2 \in \{0.10, 0.15, 0.20\}$, for the DSZ study treatment group [2], Zelenko's complete April 2020 data set [9], Zelenko's complete June 2020 data set [10], Procter's observational studies [11,12], and Raoult's high-risk (older than 60) treatment group [13]. We highlight in bold font the Bayes factors that violate the condition $b(x_0, p_2) \geq 3.7$.

| Bayes factors at the mortality rate efficacy thresholds | | | | |
|---|---|---|---|---|
| **Study** | **99.9% threshold** | **log Bayes factors** | | |
| | | $p_2 = 0.02$ | $p_2 = 0.05$ | $p_2 = 0.1$ |
| DSZ study | 7.0% | **N/A** | **N/A** | **3.51** |
| Zelenko 04/2020 | 2.9% | **N/A** | **3.47** | 4.00 |
| Zelenko 06/2020 | 1.6% | **3.43** | 4.34 | 4.73 |
| Procter I | 3.1% | **N/A** | 3.59 | 4.16 |
| Procter II | 1.4% | **3.38** | 4.15 | 4.53 |
| Raoult | 1.18% | **3.49** | 4.16 | 4.52 |
| Bayes factors at the hospitalization rate efficacy thresholds | | | | |
| **Study** | **99.9% threshold** | **log Bayes factors** | | |
| | | $p_2 = 0.10$ | $p_2 = 0.15$ | $p_2 = 0.20$ |
| DSZ study | 10.6% | **N/A** | **3.17** | **3.49** |
| Zelenko 04/2020 | 4.7% | **3.68** | 3.97 | 4.15 |
| Zelenko 06/2020 | 3.5% | 3.75 | 4.00 | 4.16 |
| Procter I | 5.9% | **3.45** | 3.80 | 3.99 |
| Procter II | 4.5% | **3.54** | 3.82 | 3.99 |

**Table 11.** Comparison of the 99% confidence efficacy threshold (Sterne interval [71]) for mortality and hospitalization rate reduction with the Bayes factor efficacy thresholds at log Bayes = 2.7, using maximum untreated mortality rate $p_2$ for high-risk patients at $p_2 \in \{0.02, 0.05, 0.10\}$ and maximum untreated hospitalization rate $p_2$ for high-risk patients at $p_2 \in \{0.10, 0.15, 0.20\}$, for the DSZ study treatment group [2], Zelenko's complete April 2020 data set [9], Zelenko's complete June 2020 data set [10], Procter's observational studies [11,12], and Raoult's high-risk (older than 60) treatment group [13]. We highlight in bold font the Bayesian thresholds that exceed the frequentist thresholds.

| Mortality rate Bayesian efficacy thresholds | | | | |
|---|---|---|---|---|
| **Study** | **99% threshold** | **log Bayes = 2.7 thresholds** | | |
| | | $p_2 = 2\%$ | $p_2 = 5\%$ | $p_2 = 10\%$ |
| DSZ study | 5.3% | **N/A** | **N/A** | 5.3% |
| Zelenko 04/2020 | 2.4% | **N/A** | 2.4% | 2.0% |
| Zelenko 06/2020 | 1.2% | **1.3%** | 1.1% | 0.9% |
| Procter I | 2.3% | **N/A** | 2.3% | 1.9% |
| Procter II | 1.08% | **1.14%** | 0.92% | 0.80% |
| Raoult | 0.96% | **1.0%** | 0.86% | 0.77% |
| Hospitalization rate Bayesian efficacy thresholds | | | | |
| **Study** | **99% threshold** | **log Bayes = 2.7 thresholds** | | |
| | | $p_2 = 10\%$ | $p_2 = 15\%$ | $p_2 = 20\%$ |
| DSZ study | 8.8% | **N/A** | **9.5%** | **8.9%** |
| Zelenko 04/2020 | 3.9% | **N/A** | 3.7% | 3.5% |
| Zelenko 06/2020 | 3.0% | 3.0% | 2.9% | 2.8% |
| Procter I | 4.9% | **5.1%** | 4.8% | 4.6% |
| Procter II | 4.0% | **4.1%** | 3.9% | 3.8% |

**Table 12.** Comparison of the 99.9% confidence efficacy threshold (Sterne interval [71]) for mortality and hospitalization rate reduction with the Bayes factor efficacy thresholds at log Bayes = 3.7, using maximum untreated mortality rate $p_2$ for high-risk patients at $p_2 \in \{0.02, 0.05, 0.10\}$ and maximum untreated hospitalization rate $p_2$ for high-risk patients at $p_2 \in \{0.10, 0.15, 0.20\}$, for the DSZ study treatment group [2], Zelenko's complete April 2020 data set [9], Zelenko's complete June 2020 data set [10], Procter's observational studies [11,12], and Raoult's high-risk (older than 60) treatment group [13]. We highlight in bold font the Bayesian thresholds that exceed the frequentist thresholds.

| **Mortality rate Bayesian efficacy thresholds** | | | | |
|---|---|---|---|---|
| **Study** | **99.9% threshold** | **log Bayes = 3.7 thresholds** | | |
| | | $p_2 = 2\%$ | $p_2 = 5\%$ | $p_2 = 10\%$ |
| DSZ study | 7.0% | **N/A** | **N/A** | **7.4%** |
| Zelenko 04/2020 | 2.9% | **N/A** | **3.1%** | 2.7% |
| Zelenko 06/2020 | 1.6% | **1.8%** | 1.4% | 1.3% |
| Procter I | 3.1% | **N/A** | **3.2%** | 2.8% |
| Procter II | 1.4% | **1.53%** | 1.26% | 1.14% |
| Raoult | 1.18% | **1.23%** | 1.08% | 1.01% |
| **Hospitalization rate Bayesian efficacy thresholds** | | | | |
| **Study** | **99.9% threshold** | **log Bayes = 3.7 thresholds** | | |
| | | $p_2 = 10\%$ | $p_2 = 15\%$ | $p_2 = 20\%$ |
| DSZ study | 10.6% | **N/A** | **11.9%** | **11.1%** |
| Zelenko 04/2020 | 4.7% | **4.8%** | 4.5% | 4.4% |
| Zelenko 06/2020 | 3.5% | 3.5% | 3.4% | 3.3% |
| Procter I | 5.9% | **6.2%** | 5.8% | 5.7% |
| Procter II | 4.5% | **4.6%** | 4.5% | 4.4% |

## 5. Discussion and Conclusions

Our findings fully support risk stratification in the management of acute COVID-19, with the intent of reducing the intensity and duration of symptoms and by that mechanism, lower the risk of hospitalization and death. Although COVID-19 is generally known as a respiratory disease, there is an accumulation of evidence [42,97,98] that it is also, if not primarily, a vascular disease, with endothelial injury having a major role in sustained permanent injuries, hospitalizations, and death. The spike protein has been shown to damage the vascular endothelial cells [42] by downregulating ACE2, thereby inhibiting mitochondrial function, and by impairing the bioavailability of nitric oxide to endothelial cells. The spike protein also triggers immune dysregulation, triggering endothelial cells to transition to an activated immune response state, which causes both macrovascular and diffuse microvascular thrombosis, leading to myocardial injury and other organ damage [97,98]. Early outpatient treatment, using multiple drugs in combination, prevents these adverse outcomes by stopping viral replication at the first phase of the illness, and mitigating the injuries caused by the hyper inflammatory COVID-19 pneumonia phase and the subsequent thromboembolic phase.

One of the lessons learned during the COVID-19 pandemic is that some of the key discoveries for the successful treatment of a novel disease emerge from the experience of the frontline doctors that are directly confronted with the need to find a way to help their patients. A conceptual understanding of the biological mechanisms via which a disease agent infects and harms patients can be used to rapidly identify therapeutic strategies, based on repurposed drugs, that may counter the disease and its sequelae. In the absence of proven and effective treatment protocols, physicians have a *duty to treat*, with informed consent from their patients, requiring an effort to innovate and/or adopt such novel therapeutic strategies, in order to immediately reduce hospitalizations and deaths and to alleviate suffering [66]. Although the orthodox approach is to consider possible treatments as unproven until they are validated with an RCT, in real life, it is possible to be confronted with a situation where the real-world observational data that result from clinical practice

are sufficiently strong to justify the immediate adoption of a newly discovered treatment protocol, and to raise the ethical concern of whether it is appropriate to even conduct the RCT, and deny treatment to a very large cohort of patients, in order to form a control group [18]. Consequently, there is a need to be able to analyze the quality of observational data in a statistically rigorous way.

We have provided a hybrid statistical framework for assessing observational evidence that combines both frequentist and Bayesian methods; the frequentist methods aim to control the *p*-value for rejecting the null hypothesis, whereas the Bayesian methods aim to control the false positive rate. The two methods are complementary and not mutually exclusive. We have also proposed a formalism for assessing the signal of efficacy with respect to both random and systemic selection bias, and explain how it can be integrated with the proposed hybrid frequentist–Bayesian method. We stress that the method aims to answer only the question of whether we are confident that the proposed treatment protocol works, in order to facilitate the binary choice of whether or not it should be adopted. An exact measurement of the efficacy is not our primary concern; we only need to establish positive as opposed to null or negative efficacy.

The main weakness of the proposed statistical methodology is that it has to be limited only to the assessment of treatments that are based on repurposed medications [17] with known acceptable safety. It would be highly inappropriate to use this approach on new medications, or other countermeasures, where the balance of risks and benefits is yet to be determined. Furthermore, the analysis of the treatment group case series needs to be compared with a model that can, at minimum, lower-bound the probability of adverse outcomes without treatment, based on our prior knowledge. On the other hand, the development of this model can be done independently from the analysis of the treatment group case series.

One way in which our approach deviates from the usual way of doing things is that we are using the proposed statistical methodology to assess the efficacy of the entire treatment algorithm against supportive care. Both the original Zelenko protocol [2] and the more enhanced McCullough protocol [14–16] are examples of sequenced multidrug treatment protocols. Furthermore, both protocols are algorithmic, in the sense that treatment is customized to the individual patient based on the patient's medical history and the response to treatment. For the case of the Zelenko protocol [2] this is done via the risk stratification of patients to low-risk and high-risk patients. For the case of the McCullough protocol [14–16], this is done both by risk stratification and also by accounting for the progression of the illness through the three distinct stages and response to treatment. Consequently, the immediate goal is not to establish that any one particular drug is effective. The goal is to establish that the treatment algorithm itself is effective, so that it can be deployed rapidly on an emergency basis and be subsequently improved over time with further research.

A possible theoretical criticism is that the particular case series that we have analyzed may have selection bias. This is mitigated, to some extent, by having reported case series from three different treatment centers, two in the United States and one in France, with consistent mortality rates; this consistency is compelling statistical evidence against geographic selection bias. More importantly, for both of the Zelenko [2,9,10] and Procter [11,12] case series, we have two consecutive reports over two consecutive time intervals replicating the hospitalization and mortality rate reduction outcomes, and these replications are additional statistical evidence against reporting selection bias. Furthermore, the treatment protocols have known biological mechanisms of action that have been reviewed in Section 1. Finally, we have introduced the idea of random selection bias thresholds that can be used to account for random selection bias. For the Zelenko June 2020 [10], Procter II [12], and Raoult [13] case series, we can have 95% confidence that random selection bias cannot be entirely responsible for the positive signal of benefit in mortality and hospitalization rate reduction. Furthermore, for the Raoult case series [13], systemic selection bias that favors the selection of high-risk healthy patients by a factor of up to 2.55 (a conservative estimate) is not sufficient to overturn the positive signal of efficacy.

The case series that we have analyzed in this paper add up to a total of 3164 high-risk patients. It is currently estimated that the total number of high-risk patients that have been treated with early outpatient treatment protocols throughout the United States may exceed this number by one or two orders of magnitude [99]. Unfortunately, no resources have been allocated to study this data by our public health agencies, but we can make some suggestions about how such an analysis could be carried out. One idea for quickly analyzing a very large data set is to extract the age $> 50$ and/or age $> 65$ part of the database, calculate the corresponding efficacy thresholds for hospitalization rate reduction and mortality rate reduction, and compare them with the CDC estimates [62–64] for number of hospitalizations and deaths for these age groups over the total number of cases with symptomatic illness. Given a large enough data set, it would also be interesting to further risk stratify the age $> 50$ and/or age $> 65$ cohorts with respect to number of days between initial symptoms and initiation of treatment, and calculate the efficacy thresholds as a function of the delay in initiating treatment. This analysis would inadvertently not include younger patients that are high risk due to comorbidities or shortness of breath presentation, however, it has the advantage that it can be carried out quickly with limited resources.

Furthermore, it would be useful to break down the case series data in sequential time intervals corresponding to different waves and different variants of the SARS-CoV-2 (Severe Acute Respiratory Syndrome Coronavirus 2) virus. The case series considered in this paper are limited to 2020, before the vaccine roll out, during which natural immunity held up at preventing reinfection [100] up until the emergence of the omicron variants near the end of 2021, which broke through natural immunity from previous variants, but also provided back immunity to the delta variant [101]. Nevertheless, in terms of general methodology, it would also be useful to subject any results to sensitivity analysis with respect to host immunity (i.e., history of previous infection and or vaccination status), as needed. Analyzing the data from several more treatment centers that have adopted early outpatient treatment protocols for high-risk patients would further mitigate the potential for selection bias.

With substantial resources, a more detailed analysis, based on the virtual control group methodology [18], is possible that can consider the entire data set and actually estimate the treatment efficacy. Given a case series of $N$ patients, one can input the medical history of each patient to the Cleveland Clinic calculator [95] and use their mathematical model to predict the probability of hospitalization and death for each patient individually. Knowing the corresponding sequence of probabilities $\mathbf{q} = (p_1, p_2, \ldots, p_N)$ for an adverse outcome (hospitalization or death) for all patients, the probability $\mathrm{pr}(N, a|\mathbf{q})$ of seeing $a$ adverse outcomes follows a Poisson binomial distribution [102], and it can be substituted to Equation (2) in order to calculate the $p$-value for rejecting the null hypothesis of no treatment efficacy. Because the probability of an adverse outcome is known for each patient, note that there is no need to worry about selection bias or calculating any efficacy thresholds, and it is possible instead to directly calculate the $p$-value for rejecting the null hypothesis.. Furthermore, since the mean of the Poisson binomial distribution is the average $q = (1/N)(p_1 + p_2 + \ldots + p_n)$ of the individual probabilities, one can calculate the risk ratio via the equation $\mathrm{RR} = a/(qN)$. To conduct the corresponding Bayesian analysis, we can assume that the effect of the early outpatient treatment is to reduce the probabilities of adverse outcome by a numerical factor $x$ to $x\mathbf{q} = (xp_1, xp_2, \ldots, xp_N)$ with $0 \leq x \leq 1$ and use the Poisson binomial distribution $\mathrm{pr}(N, a|x\mathbf{q})$ in Equation (29) and Equation (32) to calculate the corresponding integrals needed for the Bayesian factor. All other aspects of the Bayesian analysis would remain the same, except that the hypothesis being validated would not concern any efficacy thresholds, but would instead concern hypotheses about the actual efficacy $x$ of the early outpatient treatment protocol.

That said, we do not mean to imply that such a detailed analysis is necessary in order to greenlight the use of the investigated early outpatient treatment protocols for COVID-19. However, we wish to highlight that such a detailed analysis is indeed possible to carry out, using existing data and prior mathematical modeling, in order to validate the McCullough

protocol. A limitation of the Cleveland Clinic calculator is that it should ideally be used in conjunction with case series over time intervals that are aligned with the data set used to train the calculator's mathematical predictive model. Because the Cleveland Clinic calculator used data collected between 4 March 2020 and 14 July 2020 it can certainly be applied to case series up until July 2020. However, we believe that it can also be extended up until and including the delta variant, which became dominant towards the end of 2021, since all of these subsequent variants were just as hard to treat or harder than the initial waves in 2020.

Notwithstanding the hesitancy confronting the adoption of early treatment protocols for COVID-19 [94,103], everything that we have been through during the last two years vindicates the position of Frieden [54] that there is an urgent need to leverage and overcome the limitations of real-world evidence data, in order to deploy a timely life-saving response to urgent health issues. Although case series real-world data is viewed as imperfect from an epidemiological viewpoint, this viewpoint is predicated on the goal being the *unbiased measurement* of treatment efficacy. We have explained how case series data of high-risk patients for a treatment protocol based on repurposed medications, combined with our prior knowledge of population-level probabilities for adverse binary outcomes, can be used to answer the simpler question of whether or not the treatment protocol actually works (i.e., showing only the *existence* of efficacy), in order to make the up or down decision about whether or not to adopt it. The proposed statistical framework also provides a rigorous technique for quantifying the quality of these data. This can help to make objective policies on the appropriate thresholds for adopting such treatments as a standard of care. There is still an opportunity to learn much by analyzing data from various treatment centers here in the United States that treated COVID-19 with early outpatient treatment protocols, as well as treatment centers from all around the world. It is also necessary to reflect on and develop policies and procedures for leveraging the direct experience of frontline doctors treating patients towards an agile and effective response to future epidemics and pandemics.

**Supplementary Materials:** The following supporting information can be downloaded at: https://www.mdpi.com/article/10.3390/covid2080084/s1.

**Author Contributions:** Conceptualization, E.G., P.A.M., and V.Z.; methodology, E.G.; software, E.G.; validation, E.G.; formal analysis, E.G.; investigation, E.G., P.A.M., and V.Z.; writing—original draft, E.G.; writing—review and editing, E.G., P.A.M., and V.Z.; visualization, E.G. Our co-author V.Z. passed away on 30 July 2022; he has read and agreed to the 6 June 2022 preprint [104] version of this manuscript, which has since undegone minor revision before acceptance for publication, without editing of the paragraphs contributed by V.Z. All authors have read and agreed to the published version of the manuscript.

**Funding:** This research received no external funding.

**Institutional Review Board Statement:** Not applicable. This study is a reanalysis of previously published data.

**Informed Consent Statement:** Not applicable. This study is a reanalysis of previously published data.

**Data Availability Statement:** The computer code and the detailed calculations for the results reported in this paper are available via the supplementary data document [27].

**Acknowledgments:** It is a pleasure to thank Roland Derwand, Harvey Risch, and Marc Rendell for correspondence and encouragement. In particular, we wish to thank Roland Derwand for bringing the Israeli study [61] to our attention, and Harvey Risch for highlighting the importance of the papers by Frieden [54] and Deaton and Cartwright [55] and for the suggestion that we look into the coverage probability, at an early stage of our work. We also thank Marc Rendell for suggesting that we introduce a quantitative technique to mitigate selection bias, and an anonymous reviewer for very insightful comments on our manuscript throughout its development. Finally, we wish to acknowledge Lawrence Huntoon for encouraging us to look into a Bayesian approach at a very early stage of this research project.

**Conflicts of Interest:** The authors declare no conflict of interest.

## Abbreviations

The following abbreviations are used in this manuscript:

| | |
|---|---|
| SARS-CoV-2 | Severe Acute Respiratory Syndrome Coronavirus 2 |
| RCT | Randomized Controlled Trial |
| RDRP | RNA Dependent RNA Polymerase |
| EGCG | Epigallocatechin Gallate |
| RSV | Respiratory Syncytial Virus |
| DSZ | Derwand–Scholz–Zelenko |
| PCR | Polymerase Chain Reaction |
| IgG | Immunoglobulin G |
| BMI | Body Mass Index |
| CDC | Centers for Disease Control and Prevention |
| TNF-$\alpha$ | Tumor Necrosis Factor Alpha |
| IL-1$\beta$ | Interleukin-1-Beta |
| IL-6 | Interleukin 6 |

## Appendix A. Exact Fisher Test in the Limit of an Infinite Control Group

Let $N$ be the total number of patients in the treatment group, let $a$ be the number of patients with an adverse outcome (hospitalization or death) in the treatment group, let $M$ be the total number of patients in the control group, and let $b$ be the number of patients in the control group with an adverse outcome. In this appendix we will show that in the limit of an infinite control group $(M, b)$ with $x = b/M$, the $p$-value $p(N, a, M, b)$ obtained from the two-tail exact Fisher test converges to $p(N, a, x)$.

In the exact Fisher test, we assume that $N$, $M$, and $a + b$, are fixed numbers, and under the null hypothesis, we also assume that the distribution of the total $a + b$ patients with an adverse outcome between the treatment group and control group is random, with equal probability for every possible combination. It follows that under the null hypothesis, the probability of seeing a particular event $(N, a, M, b)$ is given by

$$\mathrm{pr}(N, a, M, b) = \frac{\binom{a+b}{b}\binom{N+M-a-b}{N-a}}{\binom{N+M}{N}}. \tag{A1}$$

The corresponding $p$-value is the probability of observing the event $(N, a, M, b)$ or any other less probable event, and it is given by

$$p(N, a, M, b) = \sum_{n=0}^{\min\{N, a+b\}} \mathrm{pr}(N, n, M, a+b-n)H(\mathrm{pr}(N, a, M, b) - \mathrm{pr}(N, n, M, a+b-n)), \tag{A2}$$

We note that the summation variable $n$ is restricted by both the total size $N$ of the treatment group and the total number $a + b$ of the patients with an adverse outcome, so the permissible range for all possible events is $0 \leq n \leq \min\{N, a+b\}$.

A key insight is that in the definition of $\mathrm{pr}(N, a, M, b)$, the variable $M$ can be replaced with a continuous real number, because it appears only in the top argument of the corresponding binomial coefficients. Recall that for all $a \in \mathbb{R}$ and $n \in \mathbb{N}$ the extended definition of the binomial coefficient is given by

$$\binom{a}{n} = \frac{1}{n!}\prod_{\lambda=1}^{n}(a+1-\lambda) = \frac{1}{n!}\prod_{\lambda=1}^{n}(a+1-(n-\lambda+1)) = \frac{1}{n!}\prod_{\lambda=1}^{n}(a-n+\lambda). \tag{A3}$$

On the second step, we have used the transformation $\lambda \mapsto n - \lambda + 1$, which reverses the order of factors in the product. It follows that for all $M \in \mathbb{R}$, the corresponding $M$-dependent binomial coefficients are given by

$$\binom{N + M - a - b}{N - a} = \frac{1}{(N - a)!} \prod_{\lambda=1}^{N-a} ((N + M - a - b) - (N - a) + \lambda) \tag{A4}$$

$$= \frac{1}{(N - a)!} \prod_{\lambda=1}^{N-a} (M - b + \lambda), \tag{A5}$$

and

$$\binom{N + M}{N} = \frac{1}{N!} \prod_{\lambda=1}^{N} ((N + M) - N + \lambda) = \frac{1}{N!} \prod_{\lambda=1}^{N} (M + \lambda), \tag{A6}$$

and thus, the hypergeometric probability distribution $\mathrm{pr}(N, a, M, b)$ can be rewritten as

$$\mathrm{pr}(N, a, M, b) = \frac{\binom{a + b}{b} \binom{N + M - a - b}{N - a}}{\binom{N + M}{N}} \tag{A7}$$

$$= \frac{(a + b)!}{a! b!} \left[ \frac{1}{(N - a)!} \prod_{\lambda=1}^{N-a} (M - b + \lambda) \right] \left[ N! \prod_{\lambda=1}^{N} \left( \frac{1}{M + \lambda} \right) \right] \tag{A8}$$

$$= \frac{N!}{a!(N - a)!} \frac{(a + b)!}{b!} \prod_{\lambda=1}^{N-a} (M - b + \lambda) \prod_{\lambda=1}^{N} \left( \frac{1}{M + \lambda} \right) \tag{A9}$$

$$= \binom{N}{a} \prod_{\lambda=1}^{a} (b + \lambda) \prod_{\lambda=1}^{N-a} (M - b + \lambda) \prod_{\lambda=1}^{a} \left( \frac{1}{M + \lambda} \right) \prod_{\lambda=1}^{N-a} \left( \frac{1}{M + a + gl} \right) \tag{A10}$$

$$= \binom{N}{a} \prod_{\lambda=1}^{a} \left( \frac{b + \lambda}{M + \lambda} \right) \prod_{\lambda=1}^{N-a} \left( \frac{M - b + \lambda}{M + a + \lambda} \right). \tag{A11}$$

To take the limit of an infinite control group with probability $x$ of an adverse outcome, we set $b = xM$, or equivalently $M = (1/x)b$, and take a sequence limit $b \in \mathbb{N}$ to infinity. We conclude that

$$\lim_{b \in \mathbb{N}} \mathrm{pr}(N, a, (1/x)b, b) = \binom{N}{a} \left[ \prod_{\lambda=1}^{a} \lim_{b \in \mathbb{N}} \left( \frac{b + \lambda}{(1/x)b + \lambda} \right) \right] \left[ \prod_{\lambda=1}^{N-a} \lim_{b \in \mathbb{N}} \left( \frac{(1/x)b - b + \lambda}{(1/x)b + a + \lambda} \right) \right] \tag{A12}$$

$$= \binom{N}{a} \left( \frac{1}{1/x} \right)^a \left( \frac{1/x - 1}{1/x} \right)^{N-a} \tag{A13}$$

$$= \binom{N}{a} x^a (1 - x)^{N-a} = \mathrm{pr}(N, a | x). \tag{A14}$$

An immediate consequence is that the corresponding $p$-values satisfy a similar relationship that reads

$$\lim_{b \in \mathbb{N}^*} p(N, a, (1/x)b, b) = p(N, a | x). \tag{A15}$$

The probability sums on both sides of Equation (A15) involve a variable $n$ that goes from 0 to $N$, making the number of terms on the left-hand-side probability sum independent of the size of the control group, as soon as $b$ is large enough. This makes it possible to derive Equation (A15) as an immediate consequence of Equation (A14).

**Appendix B. Calculation of the Selection Bias Thresholds**

We provide the mathematical justification for the calculation of the random selection bias threshold given by Equations (7) and (8). Suppose that we have a case series $(N, a)$ of $N$ treated patients with $a$ adverse outcomes, and have already calculated an appropriate efficacy threshold $x_0$, using the techniques detailed in Sections 2 and 3. Let $x$ be the corresponding probability of an adverse event, in the absence of treatment, as has been observed at the population level, under the same selection criteria for high-risk patient classification, as used for forming the treated case series.

The true rate $x'$ of an adverse event for the selected $N$ patients, had they not received treatment, could range from a minimum value $m_1(N, x, p_0)/N$ to a maximum value $m_2(N, x, p_0)/N$, and we would like to be able to assert with $1 - p_0$ confidence that $m_1(N, x, p_0)/N \leq x' \leq m_2(N, x, p_0)/N$. The problem of determining $m_1$ and $m_2$ is the "mirror image" of the binomial proportion confidence interval problem, reviewed in Section 2.2. With the latter case, we have a known observed event, and seek a confidence interval for the probability that generated the event. Here, we have a given probability $x$, and need the confidence interval for the number of adverse events that we expect to see in the finite sample of $N$ patients chosen out of the general population.

To calculate the confidence interval for $x'$, we consider the inequality

$$p(N, m, x) \geq p_0, \tag{A16}$$

and let $m_1(N, x, p_0)$ be the minimum natural number and $m_2(N, x, p_0)$ be the maximum natural number for the parameter $m$ that satisfies Equation (A16). We note that both of these numbers are dependent on the sample size $N$, the probability $x$ of an adverse outcome without treatment, and the $p$-value threshold $p_0$ for establishing statistical confidence. Let $S(N, x, p_0)$ be the set of all natural numbers between $m_1(N, x, p_0)$ and $m_2(N, x, p_0)$, also including $m_1(N, x, p_0)$ and $m_2(N, x, p_0)$. Let $S_0(N, x, p_0)$ be the set of all $m$ that satisfy Equation (A16). Because $p(N, m, x)$ is not monotonic with respect to $x$, the solution set $S_0(N, x, p_0)$ could be punctuated with empty gaps, and therefore we expect that $S_0(N, x, p_0) \subseteq S(N, x, p_0)$. We also note that Equation (A16) defines the indicator function $I(N, n, x, p_0)$ for the Sterne interval solution [71] of the binomial proportion confidence interval problem, which is given by

$$I(N, n, x, p_0) = \begin{cases} 1, & \text{if } p(N, n, x) \geq p_0 \\ 0, & \text{if } p(N, n, x) < p_0, \end{cases} \tag{A17}$$

and proceed with the assumption that Sterne interval [71] has conservative coverage probability.

From the above, we conclude that the probability $p(N, p_0|x)$ for seeing an outcome of $n$ adverse events with $m_1(N, x, p_0) \leq n \leq m_2(N, x, p_0)$, in the absence of treatment, for a cohort of $N$ patients, that are equivalent in every respect to the selected $N$ patients that did receive treatment in our treatment group case series, satisfies,

$$p(N, p_0|x) = \sum_{n \in S(N, x, p_0)} \text{pr}(N, n|x) \tag{A18}$$

$$\geq \sum_{n \in S_0(N, x, p_0)} \text{pr}(N, n|x) \tag{A19}$$

$$= \sum_{n=0}^{N} I(N, n, x, p_0)\text{pr}(N, n|x) \tag{A20}$$

$$= c(N, p_0|x) \geq 1 - p_0. \tag{A21}$$

The first inequality step follows from $S_0(N, x, p_0) \subseteq S(N, x, p_0)$. The next step follows from Equation (A17), then we apply the definition of the coverage probability, and the last inequality is based on the assumption that the Sterne interval has conservative coverage.

We conclude, therefore, that there is at least $1 - p_0$ probability that the number $n$ of adverse events without treatment would have been in the interval $m_1(N, x, p_0) \leq n \leq m_2(N, x, p_0)$ for an equivalent cohort of $N$ patients, chosen randomly out of the general population.

Now let us consider the random selection bias threshold, which is calculated as the minimum number $x_1$ that satisfies the implication

$$x > x_1(N, x_0, p_0) \implies p(N, \lceil x_0 N \rceil, x) < p_0, \tag{A22}$$

with $\lceil x_0 N \rceil$ the natural number obtained if we round up the number $x_0 N$. To show that this threshold works, we need to show that $x > x_1(N, x_0, p_0)$ implies that $x_0 < m_1(N, x, p_0)/N$, which is easily done with the following argument: First, we see that Equation (A22) implies that the number $\lceil x_0 N \rceil$ is outside of the set $S(N, x, p_0)$, which in turn means that either $\lceil x_0 N \rceil < m_1(N, x, p_0)$ or $\lceil x_0 N \rceil > m_2(N, x, p_0)$. To rule out the second possibility, we observe that if $x$ exceeds the random selection bias threshold $x_1(N, x_0, p_0)$, then it also exceeds the efficacy threshold $x_0$ and thus $x > x_0$. We also observe that the population level probability $x$ of an adverse outcome for untreated high-risk patients has to be inside its own confidence interval, i.e., $m_1(N, x, p_0)/N \leq x \leq m_2(N, x, p_0)/N$. Using these inequalities, it follows that

$$\lceil x_0 N \rceil \leq \lceil x N \rceil \leq \lceil (m_2(N, x, p_0)/N)N \rceil \tag{A23}$$
$$= \lceil m_2(N, x, p_0) \rceil = m_2(N, x, p_0), \tag{A24}$$

and therefore we can rule out $\lceil x_0 N \rceil > m_2(N, x, p_0)$. We conclude that $\lceil x_0 N \rceil < m_1(N, x, p_0)$, and therefore

$$x_0 = x_0 N / N \leq \lceil x_0 N \rceil / N < m_1(N, x, p_0)/N, \tag{A25}$$

which gives us $x_0 < m_1(N, x, p_0)/N$. Since the true rate $x'$ of an adverse event for an equivalent cohort of $N$ patients can be bound between $m_1(N, x, p_0)/N \leq x' \leq m_2(N, x, p_0)/N$ with $1 - p_0$ confidence, we can be assured that, in spite of any random selection bias, $x'$ exceeds the efficacy threshold $x_0$ with $1 - p_0$ confidence. This concludes the argument that justifies the calculation of the random selection bias threshold given by Equation (7).

To calculate the selection bias threshold $x_1(F|N, x_0, p_0)$ corresponding to systemic selection bias with factor $F$, we can simply recycle the preceding argument by considering a hypothetical population that has the systemic bias effect build into the proportions of healthy versus unhealthy patients, and then calculating the random selection bias threshold for this hypothetical population. Let $L = x/(1 - x)$ be the likelihood ratio of randomly selecting unhealthy vs. healthy patients, if there is no systemic selection bias. If there is some systemic selection bias, this ratio is modified into $L/F$. Consider a hypothetical population where the probability of an adverse outcome for high-risk patients without treatment is $\tilde{x}$ such that $L/F = \tilde{x}/(1 - \tilde{x})$. With basic algebra, we see that

$$\tilde{x} = \frac{L/F}{L/F + 1} = \frac{x}{x + F(1 - x)}. \tag{A26}$$

Selecting randomly from this hypothetical population is statistically equivalent to selecting with systemic bias $F$ from the actual population, in the sense that in both cases we obtain the same confidence interval for the probability $x'$. This equivalence implies that

$$x_1(F|N, x_0, p_0) \leq x \leq 1 \implies x_0 < m_1(N, \tilde{x}, p_0)/N. \tag{A27}$$

Furthermore, from the definition of the random selection bias threshold, we have

$$x_1(N, x_0, p_0) \leq \tilde{x} \leq 1 \implies x_0 < m_1(N, \tilde{x}, p_0)/N. \tag{A28}$$

Combining these equations, we find the systemic selection bias threshold by solving the inequality

$$x_1(N, x_0, p_0) \leq \tilde{x} \leq 1 \tag{A29}$$

$$\iff x_1(N, x_0, p_0) \leq \frac{x}{x + F(1-x)} \leq 1 \tag{A30}$$

$$\iff \frac{Fx_1(N, x_0, p_0)}{1 + (F-1)x_1(N, x_0, p_0)} \leq x \leq 1, \tag{A31}$$

which implies that the choice

$$x_1(F|N, x_0, p_0) = \frac{Fx_1(N, x_0, p_0)}{1 + (F-1)x_1(N, x_0, p_0)}, \tag{A32}$$

satisfies the definition of the systemic selection bias threshold given by Equation (A27). Furthermore, because by definition we have chosen $x_1(N, x_0, p_0)$ to be the minimum value that satisfies the implication in Equation (6), and $x_1(F|N, x_0, p_0)$ increases when $x_1(N, x_0, p_0)$ increases, for all values of $F$, it follows that the choice given by Equation (A32) is also the smallest possible choice that satisfies Equation (A27). This concludes the proof of Equation (A32).

**Appendix C. Monotonicity of the Bayesian Factor**

We prove that the function $b_0(x_0, p_2, t)$ is initially increasing and then decreasing with respect to $t$ with a maximum in the interval $[a/N, 1]$. We recall that

$$b_0(x_0, p_2, t) = \log \left[ \frac{p_2 - x_0}{t} \frac{\int_0^t x^a (1-x)^{N-a} dx}{\int_{x_0}^{p_2} x^a (1-x)^{N-a} dx} \right], \tag{A33}$$

consequently, maximizing the function $b_0(x_0, p_2, t)$ is equivalent to maximizing

$$g(t) = \frac{1}{t} \int_0^t x^a (1-x)^{N-a} \, dx, \tag{A34}$$

since all other factors are independent of $t$. For our argument, it is simpler to work with the more abstract definition

$$g(t) = \frac{1}{t} \int_0^t f(x) \, dx, \tag{A35}$$

and assume that the function $f(x)$ is increasing in the interval $[0, a/N]$, decreasing in the interval $[a/N, 1]$, and also satisfies $f(1) = 0$ and $f(x) > 0$ for all $x \in (0, 1)$. These are all general assumptions that are indeed satisfied by the binomial distribution $f(x) = x^a (1-x)^{N-a}$. Differentiating with respect to $t$ gives

$$g'(t) = \frac{-1}{t^2} \int_0^t f(x) \, dx + \frac{f(t)}{t}. \tag{A36}$$

From the assumptions $f(1) = 0$ and $f(x) > 0$ for all $x \in (0, 1)$, it immediately follows that

$$g'(1) = - \int_0^1 f(x) \, dx < 0. \tag{A37}$$

Next, we apply the integral mean-value theorem on the interval $[0, a/N]$ which requires the assumption that $f(x) > 0$ for all $x \in (0, a/N]$ and it follows that there exists $\xi \in [0, a/N]$ such that

$$f(\xi) = \frac{1}{a/N} \int_0^{a/N} f(x) \, dx. \tag{A38}$$

We use this equation to show that

$$g'(a/N) = \frac{-1}{(a/N)^2} \int_0^{a/N} f(x) \, dx + \frac{f(a/N)}{a/N} \tag{A39}$$

$$= \frac{-f(\xi)}{a/N} + \frac{f(a/N)}{a/N} \tag{A40}$$

$$= \frac{(f(a/N) - f(\xi))N}{a} > 0. \tag{A41}$$

Here, the inequality step is justified by the assumption that the function $f(x)$ is increasing in the interval $[0, a/N]$. It follows via the Bolzano theorem that there is at least one $t_0 \in [a/N, 1]$ such that $g'(t_0) = 0$, making all such $t_0$ critical points that are the possible local minimum or maximum points of $g(t)$. From Equation (A36), it follows that all such critical points $t_0$ also satisfy the equation

$$f(t_0) = \frac{1}{t_0} \int_0^{t_0} f(x) \, dx. \tag{A42}$$

We shall now use the second derivative test to show that any such critical points have to be local maxima, which in turn implies the uniqueness of only one such local maximum point in the interval $[a/N, 1]$. The second derivative of the function $g(t)$ is given by

$$g''(t) = \frac{d}{dt} \left[ \frac{-1}{t^2} \int_0^t f(x) \, dx + \frac{f(t)}{t} \right] \tag{A43}$$

$$= \frac{2}{t^3} \int_0^t f(x) \, dx - \frac{f(t)}{t^2} - \frac{f(t)}{t^2} + \frac{f'(t)}{t} \tag{A44}$$

$$= \frac{2}{t^3} \int_0^t f(x) \, dx - \frac{2f(t)}{t^2} + \frac{f'(t)}{t}, \tag{A45}$$

and for $t = t_0$, it follows that

$$g''(t_0) = \frac{2}{t_0^3} t_0 f(t_0) - \frac{2f(t_0)}{t_0^2} + \frac{f'(t_0)}{t_0} = \frac{f'(t_0)}{t_0} < 0. \tag{A46}$$

Here, the last inequality step is justified by the assumption that the function $f(x)$ is decreasing over the interval $[a/N, 1]$ and furthermore that $t_0 \in [a/N, 1]$. We conclude that all critical points in the interval $[a/N, 1]$ have to be local maxima, and by necessity this means that only one such local maximum actually exists in the interval $[a/N, 1]$. This concludes the proof of our claim.

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
