# Peer review of "Statistical Analysis Methods Applied to Early Outpatient COVID-19 Treatment Case Series Data"

_covid, doi:10.3390/covid2080084_

Round 1

Reviewer 1 Report

Dear Editor,

Thank you to give me the opportunity to review this interesting work on a new method to assess treatment efficacy using real world data for new diseases. This topic is of foremost importance. Indeed, recent pandemics have shown that RCTs are very useful but should not be the only weapon in the arsenal to respond to new deadly disease.

This work is not an anecdotal report. When confronting a new lethal disease, physicians should be very rapidly aware of the mechanisms of the disease and the putative treatment against the causes (the microbe) and the mechanism (for instance anticoagulation). Research is not restricted to a new marketed molecule, medical research should also encompass therapeutic strategies. This is particularly the case for COVID. The virus trigger a vascular disease, and early report secondarily confirmed by RCT has shown than anticoagulation improve prognosis. Physicians that have used anticoagulants as soon as they understand that microthrombosis was a pathophysiological mechanism saved more lives that physicians that waited results of randomized controlled trials.

I already know the present work since I already review this work for another journal. I report below the comment for the first MS and for the revised MS when this MS was submitted to the first journal. The present MS is much improved, focused and clear (notably figure 1 is good) and to my point of view deserve to be published.

My comment for the first version :

Dear Editor

Thank you to give me the opportunity to review this paper.

Authors are right that there is an urgent necessity to develop alternative approach to RCT for real time application by frontline clinicians, notably in specific situations of pandemic acute infectious diseases. This topic is really important and hot. It is the response to the “Insufficient evidence” that had a negative impact on care of patients when amoral knowledge and methodology are put above human.

Another important point is that RCT can be stopped by authorities. For instance, I suggest to read and cite the HYCOVID study in France (Dubée, 2021). In this study, there was about a two-fold less mortality in the group treated at day 28. However, as the expected sample size (n = 1300) was not obtained and the p-value not significant (ant death was not the main outcome), authors claimed that treatment had no effect.

Dubée V, Roy PM, Vielle B, et al. Hydroxychloroquine in mild-to-moderate coronavirus disease 2019: a placebo-controlled double blind trial. Clin Microbiol Infect. 2021;27(8):1124-1130. doi:10.1016/j.cmi.2021.03.005

Another point is the timing of treatment. Late treatment is not expected to be as effective as early treatment.

I have few other comments :

-            The MS is too long. It should be dramatically shortened and focused to better explain this valid approach to non-statistician medical doctor.

Clearly explain by a graphical abstract the approach with the counterfactual reasoning. What is expected if treatment is not provided. And how the threshold to consider the intervention effective is defined. Emphasize the importance of “prior belief”. So that Bayesian approach are particularly adequate with historical control group. 1/ you observe that a new fatal disease is emerging, 2/ you prospectively collect as soon as possible all the data and the predictive factors of death, 3/ based on your understanding of the disease and of the cause (the microbe), you try to save your patient as soon as possible.

This is consistent with the Helsinki declarations see item 37. Of the 2013 revision at Fortaleza in the JAMA (I suggest to mention it, because this support such a real world approach).

Overall, a hot topic with a relevant Bayesian approach to propose an alternative to RCT is some situation. However, the MS should be dramatically shortened and better explained and structured.

In the present form I would not accept this MS because too complicated to be easily understood by standard clinicians, but if a substantial effort is made to shorten, clarify and refocus, with a better explanation for non-statisticians person, it deserves to be reconsidered.

Morevoer, a statistician reviewer should be involved (I am not a statistician). 

Comment of the revised MS :

 Dear Editor,

From a medical point of view, I feel myself to have the skill and to be able to evaluate this MS. This work is really a great relevant timely challenge to propose a new methodology for assessing real life evidence to treat new acute highly fatal disease. Indeed, when a new disease emerge with very high lethality in specific at-risk group, medical knowledge allow to propose already available drugs based on rapid knowledge of the mechanism of the new fatal disease. As medicine remains medicine, and physiology remains physiology, any new disease is not expected to be free of all biological rules. In such context, and if any proposed treatment is effective, waiting for the results of a RCT for each therapeutic option inevitably exposes at-risk patients to an unethical and inhumane loss of opportunity. It is a denial of the knowledge and medical progress acquired over centuries.

In this context, I will be happy to write an editorial on this topic.

The revised MS is much better.

One minor comment could have been included: In infectious disease, this approach (as for RCT) should take into account both the evolution of the cause, the microbe (variant) and the host (immunity). The authors mentioned the variant but not the immunity.

From a purely statistical point of view, I do not have the skill to evaluate this MS.

As the journal is for medical doctor, I strongly encourage a graphical abstract. For instance, this picture could graphically and chronologically explain the steps : 1/ prompt observation of outcome, and identification of risk factors allowing to identify different “patient group with a specific risk” without specific treatment (at least 2 group), 2/ provide threshold expected to conclude efficacy for each of these groups, 3/ observation of outcome frequency in each predefined “patient group with a specific risk” with repurposed treatment, 4/ Comparison of frequency, 5/ decision to conclude or not that treatment is effective or not, and that evidence is sufficient or not to stop, continue study or generalize treatment.

I noted only one typo Proctor => Procter.

Final decision for the present version

This MS has been greatly improved and should be published.

Author Response

We very much appreciate the extensive feedback by Referee 1 throughout the previous revisions of our manuscript. His current comments have prompted a minor edit of the second paragraph of the Discussion and Conclusions section. We have also added an acknowledgment. The reviewer has mentioned his interest in  writing an editorial on the topic of emergency response to new diseases by use of repurposed medications. We would very much welcome an editorial on this topic.

Reviewer 2 Report

I congratulate the authors on this truly novel approach to the assessment of clinical regimens in non rct clinical research. Whilst pointing out the inefficiency of rct data for repurposed drugs in a pandemic, both because of the time constraints under emergency conditions as well as with the need to treat with therapeutic regimens rather than drugs in isolation, you offer a scientifically rigorous method to provide data of high quality for decision making at a public health level.

 This paper presents a two step statistical analytical method for case series data based on enter therapeutic regimens as opposed to the RCT which looks at single drugs in isolation. Regimens are designed using known drugs, i.e. repurposed drugs for novel pathologies, with their known safety profiles, dosages and mechanisms of action. This knowledge allows those putting the regimen together to ensure symptoms are addressed comprehensively.

The authors make it clear that their method is inappropriate for the assessment of novel drugs development.

The method requires frequentist statistical analysis for hypothesis testing followed by Bayesian factor analysis to address possible false positive rates. These statistical methods are well established, they are being applied in a novel way.

After the authors pose their method, they illustrate its use by applying it to already published clinical data. In Figure 1 the illustrate how it should be affected on the ground in the event of a pandemic.

I foresee this paper to become highly cited as it proposes a scientifically rigorous solution to the problem of evidence based medicine in a pandemic.

It’s been a privilege to review this paper. I can offer no constructive criticism for its improvement other than language edit.

Author Response

We thank the reviewer for the comments to the manuscript. We have carefully proofread  the manuscript and caught some of the needed language edits. They can be seen via the submitted DIFF file.

Reviewer 3 Report

The structure of the paper can be slightly improved by explicitly demarcate the methods and materials and the results sections. 

Apart from minor punctuation errors, the article is well written and the findings are very clearly articulated. I suggest, though, that the authors clearly demarcate the Material and Methods and the Results Section for ease of reading. I do also suggest that the authors reiterate the need to embrace uncertainty in the patient data resulting from probably the nature with which the data was collected or obtained. 

Author Response

We have restructured the manuscript, as  suggested by the reviewer. The original submitted version of the manuscript separated the discussion of the mathematical technique from the discussion of the application of the technique, which would have made the most sense for an applied statistics paper. In order to have a results section that contains only the results, it was necessary to move the material on the background of the biological mechanism of action for the Zelenko and McCullough protocols to the introduction section. We have also clearly  demarcated two distinct methods sections; one for the discussion of the frequentist technique and one for the Bayesian technique. We hope that this restructuring is what the reviewer had in mind.

With regards to the other comment about reiterating the "need to embrace uncertainty", we have done so in the last paragraph of the Conclusions and Discussion section.